# Glycolytic preconditioning in astrocytes mitigates trauma-induced neurodegeneration

Rene Solano Fonseca[1], Patrick Metang[1], Nathan Egge[1], Yingjian Liu[2], Kielen R Zuurbier[1,3], Karthigayini Sivaprakasam[3,4], Shawn Shirazi[5], Ashleigh Chuah[1], Sonja LB Arneaud[1], Genevieve Konopka[3,4], Dong Qian[2], Peter M Douglas[1,6]*

[1]Department of Molecular Biology, University of Texas Southwestern Medical Center, Dallas, United States; [2]Department of Mechanical Engineering, University of Texas at Dallas, Dallas, United States; [3]O'Donnell Brain Institute, University of Texas Southwestern Medical Center, Dallas, United States; [4]Department of Neuroscience, University of Texas Southwestern Medical Center, Dallas, United States; [5]Department of Integrative Biology, University of California, Berkeley, Berkeley, United States; [6]Hamon Center for Regenerative Science and Medicine, University of Texas Southwestern Medical Center, Dallas, United States

**Abstract** Concussion is associated with a myriad of deleterious immediate and long-term consequences. Yet the molecular mechanisms and genetic targets promoting the selective vulnerability of different neural subtypes to dysfunction and degeneration remain unclear. Translating experimental models of blunt force trauma in *C. elegans* to concussion in mice, we identify a conserved neuroprotective mechanism in which reduction of mitochondrial electron flux through complex IV suppresses trauma-induced degeneration of the highly vulnerable dopaminergic neurons. Reducing cytochrome C oxidase function elevates mitochondrial-derived reactive oxygen species, which signal through the cytosolic hypoxia inducing transcription factor, *Hif1a*, to promote hyperphosphorylation and inactivation of the pyruvate dehydrogenase, PDHE1α. This critical enzyme initiates the Warburg shunt, which drives energetic reallocation from mitochondrial respiration to astrocyte-mediated glycolysis in a neuroprotective manner. These studies demonstrate a conserved process in which glycolytic preconditioning suppresses Parkinson-like hypersensitivity of dopaminergic neurons to trauma-induced degeneration via redox signaling and the Warburg effect.

*For correspondence: peter.douglas@utsouthwestern. edu

## Introduction

Selective vulnerability of different neuronal subtypes to dysfunction, degeneration, and death underlies all neurological diseases. Differential interplay between genetic and environmental factors contribute to this neuronal selectivity. Environmental stimuli have the capacity to trigger pathogenic cascades contingent on genetic factors. Such is the case for traumatic brain injury (TBI) and concussion, in which the initial biomechanical insult precipitates numerous cellular and systemic alterations with the potential to promote neurological dysfunction and progressive neurodegeneration. In particular, ionic imbalances, cytoskeletal disruption, aberrant proteostasis, genomic instability, and mitochondrial dysfunction within neurons and support cells initiate systemic abnormalities including metabolic impairment, vascular disruption, inflammation, blood-brain barrier defects, and reduced cerebral blood-flow (*Giza et al., 2018*). Yet, the stochastic nature of the mechanical insult has

**eLife digest** Concussion is a type of traumatic brain injury that results from a sudden blow or jolt to the head. Symptoms can include a passing headache, dizziness, confusion or sensitivity to light, but experiencing multiple concussions can have drastic repercussions in later life.

Studies of professional athletes have shown that those who experience one or more concussions are prone to developing Alzheimer's and Parkinson's disease, two well-known neurodegenerative diseases. Both conditions involve the progressive loss or breakdown of nerve cells, called neurons. But exactly how this so-called neurodegeneration of brain cells stems from the original, physical injury remains unclear.

Head trauma may cause damage to the structural support of a cell or disrupt the flow of electrical impulses through neurons. Energy use and production in damaged cells could shift into overdrive to repair the damage. The chemical properties of different types of brain cells could also make some more vulnerable to trauma than others. Besides neurons, star-shaped support cells in the brain called astrocytes, which may have some protective ability, could also be affected.

To investigate which cells may be more susceptible to traumatic injuries, Solano Fonseca et al. modelled the impacts of concussion-like head trauma in roundworms (*C. elegans*) and mice. In both animals, one type of neuron was extremely vulnerable to cell death after trauma. Neurons that release dopamine, a chemical involved in cell-to-cell communication and the brain's reward system, showed signs of cell damage and deteriorated after injury. Dopaminergic cells, as these cells are called, are involved in motor coordination, and the loss of dopaminergic cells has been linked to both Alzheimer's and Parkinson's disease.

Astrocytes, however, had a role in reducing the death of dopaminergic neurons after trauma. In experiments, astrocytes appeared to restore the balance of energy production to meet the increased energy demands of impacted neurons. Single-cell analyses showed that genes involved in metabolism were switched on in astrocytes to produce energy via an alternative pathway. This energetic shift facilitated via astrocytes may help mitigate against some damage to dopamine-producing neurons after trauma, reducing cell death.

This work furthers our understanding of cellular changes in the concussed brain. More research will be required to better characterise how this immediate trauma to cells, and the subsequent loss of dopaminergic neurons, impacts brain health long-term. Efforts to design effective therapies to slow or reverse these changes could then follow.

complicated our ability to define the underlying mechanisms of neurodegeneration. It remains unclear whether the inherent chemical properties of different neuronal subtypes confer varying degrees of vulnerability to trauma-induced neurodegeneration. Additionally, the genetic contribution to this widespread disorder is poorly understood, genetic effectors are likely to amplify the cellular and systemic aberrations initiated by the mechanical insult. Thereby, we hypothesize that genetic targeting and modulation may be sufficient to suppress the ensuing neurodegeneration after concussive brain injury.

Metabolic fluctuations in the brain occurring with age or injury have been linked with neurological health (*Camandola and Mattson, 2017*; *Marino et al., 2007*). The neurometabolic cascade resulting from concussion is characterized by erratic fluctuations in metabolism, which are hypothesized to accommodate the energetic demand required for neuronal repolarization and repair (*Giza and Hovda, 2001*). Brain energetics relies heavily on two metabolic pathways including mitochondrial oxidative phosphorylation and cytosolic glycolysis (*Kasischke et al., 2004*; *Pellerin and Magistretti, 1994*). The astrocyte-neuron lactate shuttle (ANLS) entails astrocytic production and transport of the glycolytic by-product, lactate, to neurons which preferentially oxidize it rather than glucose to meet their energetic demands (*Magistretti et al., 1999*). Immediately following injury, rodent brains display signs of decreased oxidative phosphorylation that can persist for weeks (*Gilmer et al., 2009*; *Xiong et al., 1997*). Conversely, the injured brain transiently elevates normoxic glycolysis in humans and animal models (*Bergsneider et al., 1997*; *Sokoloff et al., 1977*; *Yoshino et al., 1991*), suggesting an energetic reallocation through astrocytic means. This metabolic shift is a phenomena observed in cancer cells and is termed the Warburg effect, in which aerobic glycolysis is favored

over oxidative phosphorylation (*Warburg, 1956*). Thus, the injured brain initially undergoes a Warburg-like response, but how this metabolic shift occurs within the astrocyte-neuron axis and its ability to impact neurological function and degeneration after concussion is not well understood.

## Results

### Conserved hypersensitivity of dopaminergic neurons to trauma-induced degeneration

In contrast to the cellular and network complexity of the human brain, the adult nematode, *C. elegans,* possesses 302 post-mitotic neurons (*White et al., 1986*). A simple nervous system offers the opportunity to rapidly screen different neural subtypes and gene products in pursuit of uncovering neurodegenerative mechanisms that translate to the mammalian brain. To this end, we developed a collision-based, rapid deceleration model of trauma in which high-frequency, multidirectional agitation delivers a well-calibrated injury to a large population of worms (*Egge et al., 2021*). To determine whether the inherent biochemical properties of different neural subtypes impart variable sensitivity to blunt force trauma, dopaminergic, GABAergic, glutamatergic, serotonergic, and cholinergic neurons were individually monitored using targeted GFP expression. Fluorescence retention in individual neurons housed in the central nerve ring was measured to assess viability at various time points post-trauma (*Nass et al., 2002*). When compared to age-matched non-injured counterparts, dopaminergic neurons displayed the greatest reduction in fluorescence with a 42.1% (+/−1.4%) loss in signal intensity within 24 hr after injury, indicative of elevated sensitivity to degeneration (*Figure 1A*). Microscopic examination of dopaminergic neurons revealed phenotypes characteristic of neuronal damage (*Gennarelli, 1996*; *Kilinc et al., 2009*) 24 hr after injury, including fluorescence beading in dendritic processes (*Figure 1B*).

To examine conservation of dopaminergic hypersensitivity to trauma-induced neurodegeneration, we administered a concussive close-head injury to mice. This injury impaired immediate righting reflex response, indicating temporary loss of consciousness (*Figure 1—figure supplement 1A*). Seven days after injury, dopaminergic neurons of the substantia nigra exhibited signs of increased apoptosis as evidenced by enhanced immunofluorescence staining of cleaved caspase-3, and a 59.06% (+/−3.03%) loss of tyrosine hydroxylase-positive neurons was observed 28 days post-trauma (*Figure 1C and D*; *Figure 1—figure supplement 1B–D*). Open-head models of rodent brain injury consistently report necrotic tissue loss at or near the lesion site (*Hall et al., 2005*). Yet, one study demonstrates neuronal loss within the substantia nigra (*Liu et al., 2017*). In contrast, closed-head brain injury without cranial fracture revealed no significant cleaved caspase-3 staining in the hippocampus, thalamus and visual cortex seven days post-injury (*Figure 1—figure supplement 1E–I*). Furthermore, neuroinflammation was not evident in the immunolabeling of microglia (IBA1) with CD68-positive inflammatory cytoplasmic granules or in the transcriptional activation of known inflammatory targets (*Figure 1—figure supplement 1J–K*). Dopaminergic neurons within the midbrain comprise the nigrostriatal circuit, which influences voluntary movement. In concussed mice, we observed reduced latency on the accelerating rotarod and decreased resting state ambulation coincident with dopaminergic degeneration (*Figure 1E and F*). Thus, modeling blunt force trauma in *C. elegans* and concussive injury in mice demonstrate conserved hypersensitivity of dopaminergic neurons to biomechanical insult.

To determine the physical strain on brain regions proximal and distal to the injury site, we developed a computational biophysical model of TBI. Specifications for our closed-head trauma device were incorporated into a detailed finite element model (FEM) with ~2.5 million 3D solid elements (*Figure 2—figure supplement 1A and B*). To capture stress and strain across the entire brain, six distinct material properties were assigned to accurately represent varying brain regions (*Table 1*). Biophysical modeling revealed a stress wave propagating throughout the mouse brain due to linear impact (*Figure 2—figure supplement 1C-E*; *Video 1*). By FEM analysis, a peak effective stress of 22.8 kPa (corresponding principal strain of 8.3%) first appeared in the cortex region at 0.31 ms. Then, stress propagated to 28.62 kPa (corresponding principal strain of 23.6% at the thalamus and 18.5% at the substantia nigra) at 0.85 ms post-injury in the thalamus and substantia nigra region. After impacting, a stress wave caused another peak effective stress of around 30 kPa (corresponding principal strain of 40.9% at the brainstem region and 27.0% at the thalamus region) at 1.64 ms

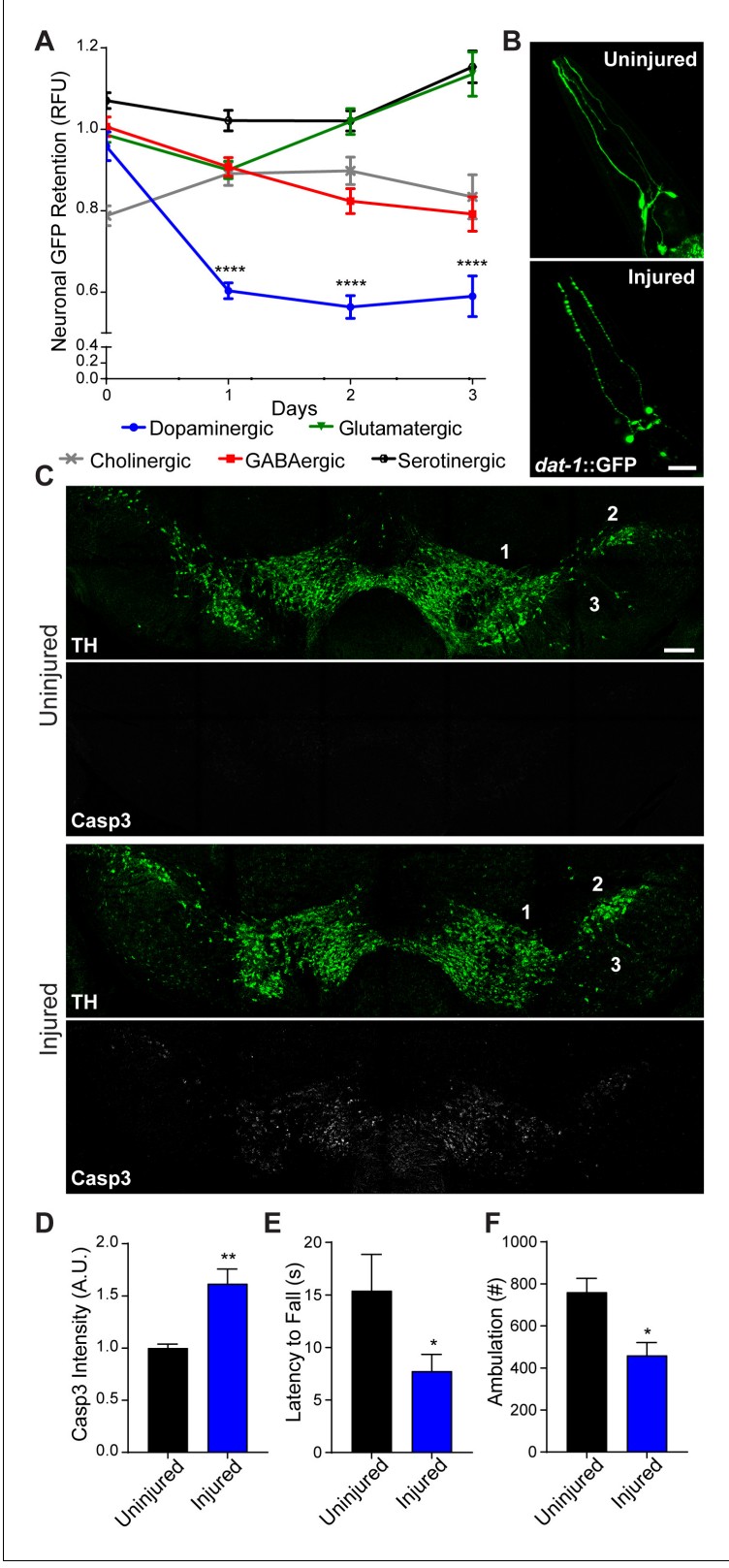

**Figure 1.** Conserved vulnerability of dopaminergic neurons to trauma-induced degeneration. (**A**) Post-trauma GFP retention in different *C. elegans* neuronal subtypes by large-particle flow cytometry. n=two sorts, n=1181 worms (dopaminergic), n=1265 worms (GABAergic), n=1115 worms (glutamatergic), n=3139 worms (serotonergic), and n=1117 worms (cholinergic). (**B**) Representative micrographs of GFP dopaminergic neuronal morphology in *C.*

*Figure 1 continued on next page*

*Figure 1 continued*

*elegans* pre-injury and 24 hr post-injury. Scale bar=20 μm. (**C**) Representative micrographs of tyrosine hydroxylase (TH; green) and cleaved caspase-3 (Casp3; gray) in the midbrain of 10 week-old mice subject to concussive injury. 1: VTA, 2: SNpc, 3: SNr. Scale bar=200 μm. (**D**) Quantification of cleaved caspase-3 staining in the midbrain. n=four uninjured, n=six injured. (**E**) Latency to fall off an accelerating rotating rod 7 days after concussive injury. n=five uninjured, n=eight injured. (**F**) Incidence of laser beam disruptions in cage per mouse over 24 hr, n=three per group. Data are mean ± SEM. *$p \leq 0.05$, **$p \leq 0.01$, ****$p \leq 0.0001$.

The online version of this article includes the following figure supplement(s) for figure 1:

**Figure supplement 1.** Immediate and long-term effects of concussive brain injury.

---

(*Figure 2A–C*; *Video 2*). Thus, dopaminergic neurons may be sensitized to biomechanical insults due to inherent physiological properties since non-invasive, linear impact generates a strain within the midbrain comparable to brain regions closer to the impact site that lack evidence of neuronal death.

## Cytochrome C oxidase deficiency protects dopaminergic neurons against trauma-induced degeneration

Longitudinal studies from the clinic show that traumatic brain injury with a loss of consciousness presents a significant risk factor for development of Parkinson's disease (PD), Parkinsonism, and Lewy body accumulation (*Crane et al., 2016*). Indeed, the pathology and motor deficits that we observed after concussive injury in mice are similar to those of individuals with PD. Based on familial links and environmental risk factors, mitochondrial components including the electron transport chain (ETC) have emerged as prominent drivers of PD (*Perier and Vila, 2012*). Utilizing our *C. elegans* trauma model, we found that reducing the complex IV cytochrome C oxidase activity of the ETC through *cox-5b* RNAi promotes a dose-dependent survival of dopaminergic neurons after injury (*Figure 3A and B*). Moreover, reduced expression of *cox-5b* exclusively in the nervous system was sufficient to promote survival of dopaminergic neurons after blunt force trauma (*Figure 3—figure supplement 1A*).

To examine the conservation of cytochrome C oxidase-related neuroprotection, we examined mice lacking the complex IV assembly factor surfeit locus protein 1, SURF1 (*Dell'agnello et al., 2007*). *Surf1* mutations reduce rather than ablate cytochrome C oxidase activity, leading to a 53.9% reduction in function (*Figure 3—figure supplement 1B*). *Surf1*[-/-] mice exhibited a 40% reduction in mortality rate following lethal head trauma and a 2.3-fold improvement in post-injury reflex recovery time compared to wild type littermates (*Figure 3—figure supplement 1C and D*), suggesting immediate protection against TBI. Cleaved caspase-3 signal intensity within the midbrain was reduced by 55% (+/−3.5%) after 7 days in concussed *Surf1*[-/-] mice (*Figure 3C and D*). Coincident with decreased neurodegeneration, *Surf1*[-/-] mice showed increased latency in high-speed rotarod analysis compared to wild-type littermates (*Figure 3E*; *Figure 3—figure supplement 1E*). This was accompanied by a 25.93% (+/−6.828%) dopaminergic loss 28 days post-injury, compared to the 59.06% (+/−3.03%) observed in wild-type counterparts (*Figure 3—figure supplement 1F–H*). Moreover, resting state ambulation was unaffected by trauma in *Surf1*[-/-] mice unlike deficits observed in wild-type littermates (*Figure 3F*). Thus, suppressing trauma-induced degeneration of dopaminergic neurons by reduced cytochrome C oxidase activity is evolutionarily conserved between *C. elegans* and mice.

**Table 1.** Viscoelastic parameters for P56 mouse brain.

| Brain region | G0 (Pa) | $g_1$ | $g_2$ | $g_l$ | $\tau_1$ (ms) | $\tau_2$ (ms) |
|---|---|---|---|---|---|---|
| Pons | 7643 | 0.578 | 0.267 | 0.162 | 12 | 182 |
| Cortex | 6343 | 0.568 | 0.264 | 0.168 | 14 | 182 |
| Cerebellum | 2807 | 0.518 | 0.31 | 0.171 | 14 | 190 |
| Thalamus | 2674 | 0.578 | 0.248 | 0.174 | 15 | 206 |
| Medulla | 3859 | 0.52 | 0.3 | 0.18 | 17 | 226 |
| Hippocampus | 5422 | 0.187 | 0.554 | 0.121 | 265 | 18 |

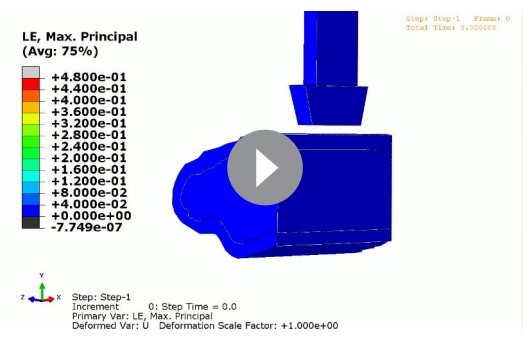

**Video 1.** Maximum principal stress propagation throughout the mouse brain after impact. https://elifesciences.org/articles/69438#video1

## Cytochrome C oxidase deficiency suppresses trauma-induced reactive oxygen species production

To understand the cause of neurodegeneration in our worm-to-mouse translational trauma model, we examined similarities reported for concussive brain injury and Parkinson's disease. Reactive oxygen species (ROS) are toxic by-products of brain injury (*Hall and Braughler, 1993*) that originate in the mitochondrial inner membrane by partial reduction of oxygen. Upon accumulation, ROS initiate cellular damage and stress. In genetic and pharmacological models of Parkinson's disease, mitochondrial dysfunction and ROS accumulation underlie dopaminergic neurodegeneration and subsequent disease progression (*Blesa et al., 2015*). In *C. elegans*, we observed a five fold increase in ROS production immediately after blunt force trauma, which was suppressed upon *cox-5b* RNAi (*Figure 4A*). Since *cox-5b* RNAi treatments in *C. elegans* are reported to reduce oxygen consumption (*Kaufman and Crowder, 2015*), we hypothesized that reduced reliance on mitochondrial respiration mitigates the likelihood of overwhelming respiratory output. Essentially, reducing the rate of electron transfer between ETC complexes reduces the rate of ROS generation and accumulation. In this model, *cox-5b* RNAi should fail to protect neurons against ectopically generated ROS. Indeed, elevating neuronal ROS via expression of the mitochondrial targeted oxidative photosensitizer, Tom20::KillerRed (*Wojtovich and Foster, 2014*), was sufficient to induce dopaminergic neurodegeneration in worms that could not be suppressed by *cox-5b* RNAi (*Figure 4B*). Thus, intra-neuronal ROS production is sufficient to drive dopaminergic degeneration, while neuroprotection conferred by reduced cytochrome C oxidase activity appears to act upstream of this ROS-induced neurotoxicity.

To evaluate the conservation of this neuroprotective mechanism in the mouse, we examined mitochondrial function and ROS production in the absence of *Surf1*. Despite no significant change in mitochondrial membrane potential (*Figure 4—figure supplement 1A*), *Surf1⁻ᐟ⁻* cells consumed two fold less oxygen and exhibited reduced reticular mitochondrial morphology (*Figure 4C and D*; *Figure 4—figure supplement 1B*), strongly suggesting reductions in oxidative phosphorylation (*Galloway et al., 2012*). Two hours after concussive injury in mice, a transient 69% increase of the ROS byproduct, superoxide, was observed in mitochondria isolated from brain tissue compared to a 6% increase in *Surf1⁻ᐟ⁻* mice (*Figure 4E*). Moreover, metabolite analysis from brain homogenates shows a 61% increase in glutathione oxidation within 2 hr of injury in wild-type animals, which was not observed in *Surf1⁻ᐟ⁻* mutants (*Figure 4F*). Thus, decreased cytochrome C oxidase activity suppresses transient trauma-induced ROS accumulation, a process conserved from worm to mouse. Notably, the basal mitochondrial superoxide levels and oxidized glutathione were significantly elevated in the *Surf1⁻ᐟ⁻* brains compared to wild-type littermates (*Figure 4—figure supplement 1C*). Therefore, minimizing transient production and accumulation of oxidative species immediately after trauma parallels neuroprotection.

## Glycolytic preconditioning by reduced cytochrome C oxidase activity protects dopaminergic neurons against concussive injury

Traumatic brain injury is a costly energetic process in which neurons work to re-establish resting membrane potential and facilitate cellular repair (*Giza and Hovda, 2001*). Despite the apparent decrease in oxidative phosphorylation (*Figure 4C*), ATP levels in the *Surf1⁻ᐟ⁻* brain were not affected compared to wild type littermates (*Figure 5—figure supplement 1A*). Furthermore, respiratory exchange rate, food intake, body temperature, and overall body weight were also unaffected in *Surf1⁻ᐟ⁻* mice (*Figure 5—figure supplement 1B–E*). Thus, *Surf1⁻ᐟ⁻* mice likely accommodate their energetic needs through alternate means in response to suboptimal mitochondrial respiration. Post-trauma, the brain transiently increases glycolysis (*Yoshino et al., 1991*) and will readily metabolize

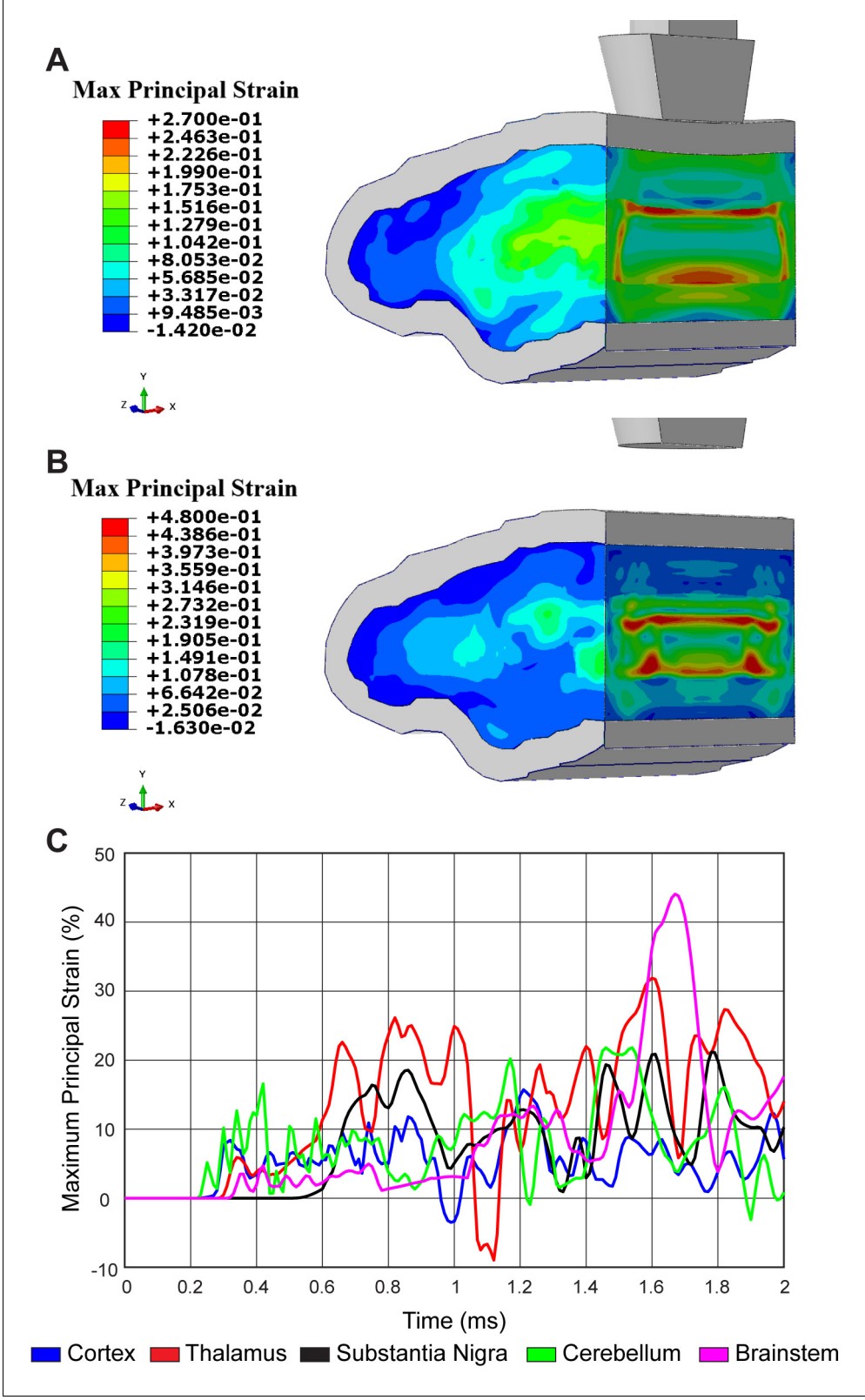

**Figure 2.** Concussive head trauma propagates strain throughout the brain. Heat map of representative brain shows predicted maximum principal strain at (**A**) 0.85 ms and (**B**) 1.64 ms after initial impact. (**C**) Time history of maximum principal strain within the cortex (blue), thalamus (red), substantia nigra (black), cerebellum (green), and brainstem (magenta).

*Figure 2 continued on next page*

*Figure 2 continued*

The online version of this article includes the following figure supplement(s) for figure 2:

**Figure supplement 1.** Finite element model of concussive head injury in the mouse.

___

the glycolytic by-product, lactate (*Glenn et al., 2015*). *Surf1*$^{-/-}$ animals appeared to undergo this Warburg-like increase in glycolysis. Consistent with elevated blood lactate reported in *Surf1*$^{-/-}$ mice (*Pulliam et al., 2014*), increased transcription of the lactate dehydrogenase, *Ldhd*, in the *Surf1*$^{-/-}$ brain correlated with the rapid acidification of growth media by *Surf1*$^{-/-}$ cells (*Figure 5A*; *Figure 5— figure supplement 1F*). Similar transcriptional upregulation of the lactate dehydrogenase, *ldh-1*, was also observed in worms treated with *cox-5b* RNAi (*Figure 5—figure supplement 1G*).

We hypothesized that neuroprotection by reduced cytochrome C oxidase is the result of preconditioning through metabolic reallocation. By preemptively shifting metabolism away from mitochondria respiration, the likelihood of overwhelming ETC function is reduced during the metabolic demand of TBI. To accommodate energetic needs, the brain relies more heavily on normoxic glycolysis in a process referred to as the Warburg effect (*Warburg, 1956*). In *Surf1*$^{-/-}$ brains, we observed transcriptional upregulation of several enzymes in the glycolytic pathway (*Figure 5A*; *Figure 5—figure supplement 1H*). As a critical regulator of the Warburg shunt, which facilitates the shift from mitochondrial respiration to cytosolic glycolysis, the pyruvate dehydrogenase complex (PDC) is inactivated by pyruvate dehydrogenase kinases, PDK1-4, through phosphorylation of its E1 alpha subunit (PDHE1α) (*Kolobova et al., 2001*; *Patel and Korotchkina, 2006*). In the brains of *Surf1* mutants, we observed transcriptional upregulation of the brain-enriched *Pdk2* in conjunction with the repression of its complementary PDH phosphatase, *Pdp1* (*Figure 5A*; *Figure 5—figure supplement 1H*). PDHE1α phosphorylation has previously been reported after open-head brain injury and provides a potential molecular explanation for trauma-induced hyperglycolysis (*Xing et al., 2009*). Consistent with transcriptional changes in *Surf1*$^{-/-}$ brains, we observed increased PDHE1α phosphorylation both before and after injury in *Surf1*$^{-/-}$ compared to wild-type littermates (*Figure 5B*; *Figure 5—figure supplement 2A*), while total PDH levels are equivalent (*Figure 5—figure supplement 2B and C*). Thus, *Surf1* mutants are preconditioned prior to injury into a Warburg-like state, which likely mitigates the transient ROS production observed immediately after concussion.

While *Surf1* mutants possess a shift in their global transcriptome, it remained unclear how this neuroprotective response occurs at the cellular level within the astrocyte-neuron axis. To obtain this cellular resolution, we employed single nuclei RNA sequencing, which allowed transcriptomic analysis of individual cells from the midbrain and striatum (*Figure 5—figure supplement 1M–O*). Consistent with the astrocyte-neuron lactate shuttle hypothesis, the brain-enriched Warburg shunt regulator, *Pdk2*, as well as every activated glycolytic gene except for *Gpi1*, was elevated in astrocytes (*Figure 5C*). Indirect immunofluorescence within the midbrain confirmed that steady-state expression of the PDK2 protein was confined to astrocytes (*Figure 5—figure supplement 1P and Q*). Moreover, glycolytic transcripts were elevated within *Surf1* mutant astrocytes compared to wild-

type littermates (*Figure 5D*). Through cellular resolution of this glycolytic preconditioning mechanism, we provide evidence that the neuroprotective Warburg shift initiated by electron transport impairment originates within astrocytes.

To understand the molecular mechanisms linking the *Surf1* mutation with the Warburg effect, we examined the hypoxia inducible factor, *Hif1a*, which promotes aerobic glycolysis by activating expression of PDK and glycolytic genes (*Kim et al., 2006*). Mitochondria act as oxygen sensors which can signal to cytosolic HIF1α through ROS production (*Guzy et al., 2005*). We confirmed that elevated ROS production caused by paraquat treatment was sufficient

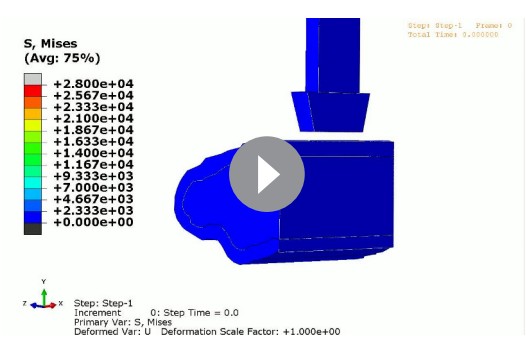

**Video 2.** Von Mises stress propagation throughout the mouse brain after impact.

https://elifesciences.org/articles/69438#video2

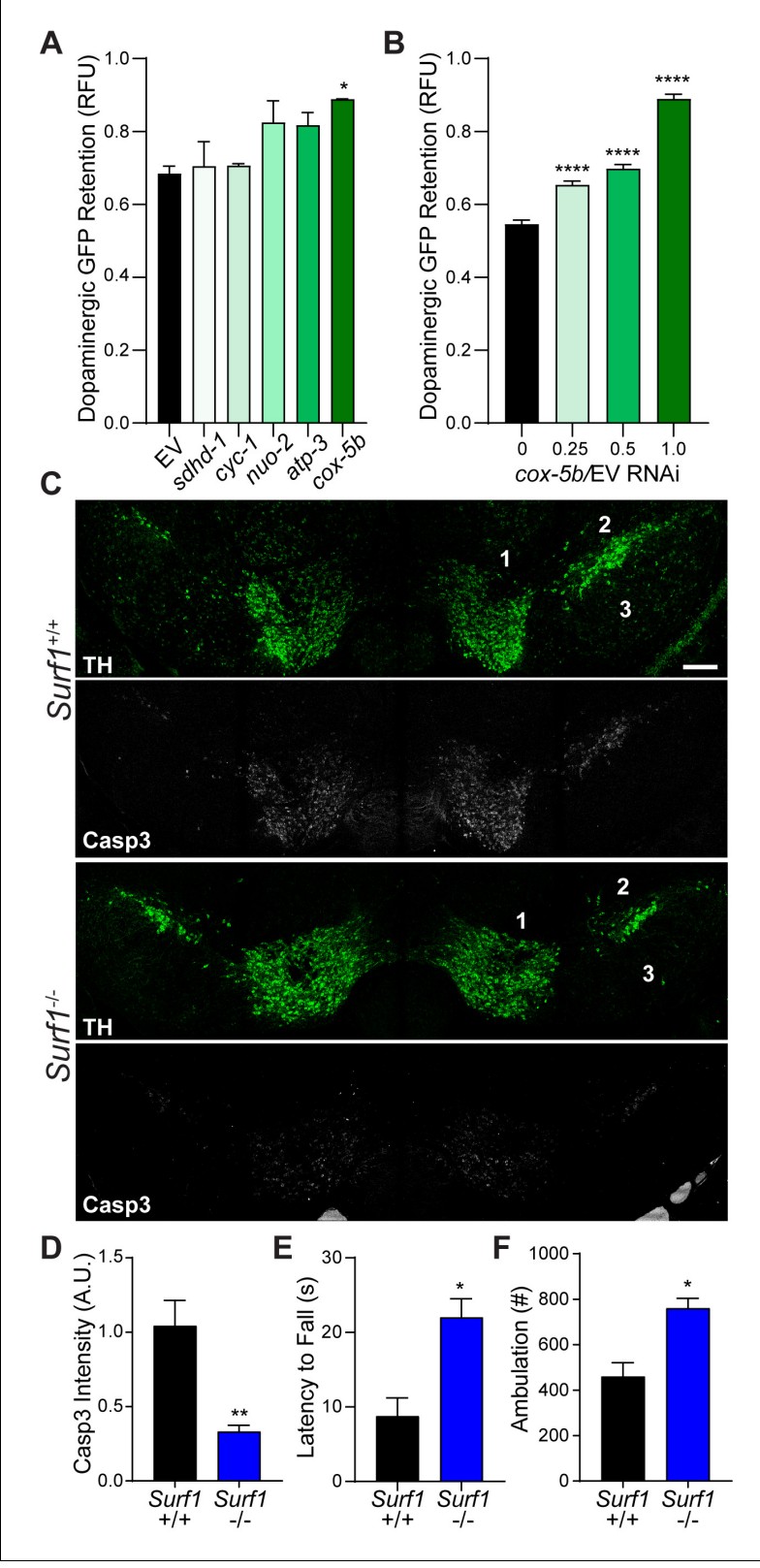

**Figure 3.** Conserved protection against trauma-induced neurodegeneration by reducing cytochrome C oxidase. (**A**) Post-trauma GFP retention in dopaminergic neurons of *C. elegans* by large-particle flow cytometry with the respective RNAi for ETC subunit components. n=two sorts, n=3344 worms (*nuo-2*), n=2343 worms (*sdhd-1*), n=3825 (*cyc-1*), n=2973 (*cox-5b*), n=3265 (*atp-3*), n=1948 (EV). (**B**) Post-trauma GFP retention in dopaminergic

*Figure 3 continued on next page*

*Figure 3 continued*

neurons of *C. elegans* treated with *cox-5b*/EV RNAi dilutions. n=six sort, n=512 worms (0), n=827 worms (0.25), n=900 worms (0.5), n=832 worms (1). (**C**) Representative micrographs of tyrosine hydroxylase (TH, green) and cleaved caspase-3 (Casp3, gray) in the midbrain of 10-week-old *Surf1*$^{+/+}$ or *Surf1*$^{-/-}$ mice subject to concussive injury. 1: VTA, 2: SNpc, 3: SNr. Scale bar=200 μm. (**D**) Quantification of cleaved caspase-3 staining in the midbrain. n=5 *Surf1*$^{+/+}$, n=6 *Surf1*$^{-/-}$. (**E**) Latency to fall off rotating rod 7 days after concussive injury (n=four per group). (**F**) Incidence of laser beam disruptions per mouse over 24 hr (n=three per group). Data are mean ± SEM. *p ≤ 0.05, **p ≤ 0.01, and ****p ≤ 0.0001.

The online version of this article includes the following figure supplement(s) for figure 3:

**Figure supplement 1.** Reduced cytochrome C oxidase impact after concussive brain injury.

to stabilize HIF1α in culture (*Figure 5—figure supplement 1J*). We hypothesized that elevated basal

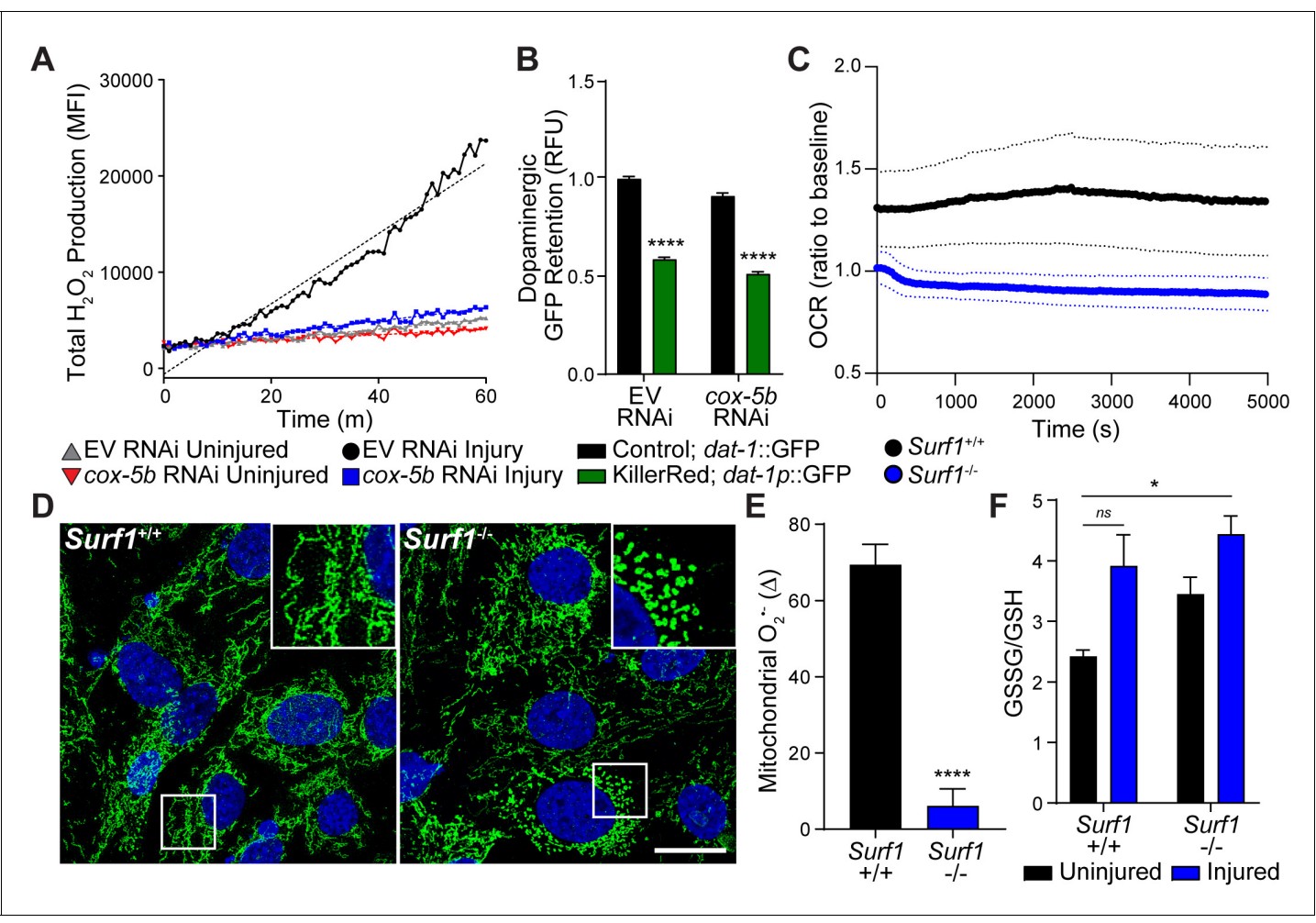

**Figure 4.** Reducing cytochrome C oxidase prevents trauma-induced ROS production. (**A**) Time-course of peroxide levels in *C. elegans* post-trauma. n=three repeats per group. (**B**) GFP retention in dopaminergic neurons of *C. elegans* ectopically expressing the mitochondrial-targeted KillerRed fluorophore in neurons. n=three repeats per group. (**C**) Oxygen consumption rate, OCR, in *Surf1*$^{+/+}$ and *Surf1*$^{-/-}$ mouse embryonic fibroblasts. n=3 *Surf1*$^{+/+}$, n=2 *Surf1*$^{-/-}$. (**D**) Representative micrographs of TOM20 (green) and DAPI (blue) in *Surf1*$^{+/+}$ and *Surf1*$^{-/-}$ mouse embryonic fibroblasts. Scale bar=20 μm. (**E**) Trauma-induced superoxide production in mitochondria from *Surf1*$^{+/+}$ and *Surf1*$^{-/-}$ brain single cell suspensions 2 hr post-injury. n=three per group. (**F**) Oxidized glutathione (GSSG) over glutathione (GSH) from metabolite analysis of *Surf1*$^{+/+}$ and *Surf1*$^{-/-}$ brain tissue subject to concussive injury. n=three per group. Data are mean ± SEM. *p ≤ 0.0116, ns (not significant, p = 0.0524) and ****p ≤ 0.0001.

The online version of this article includes the following figure supplement(s) for figure 4:

**Figure supplement 1.** Mitochondrial dynamics in embryonic fibroblasts.

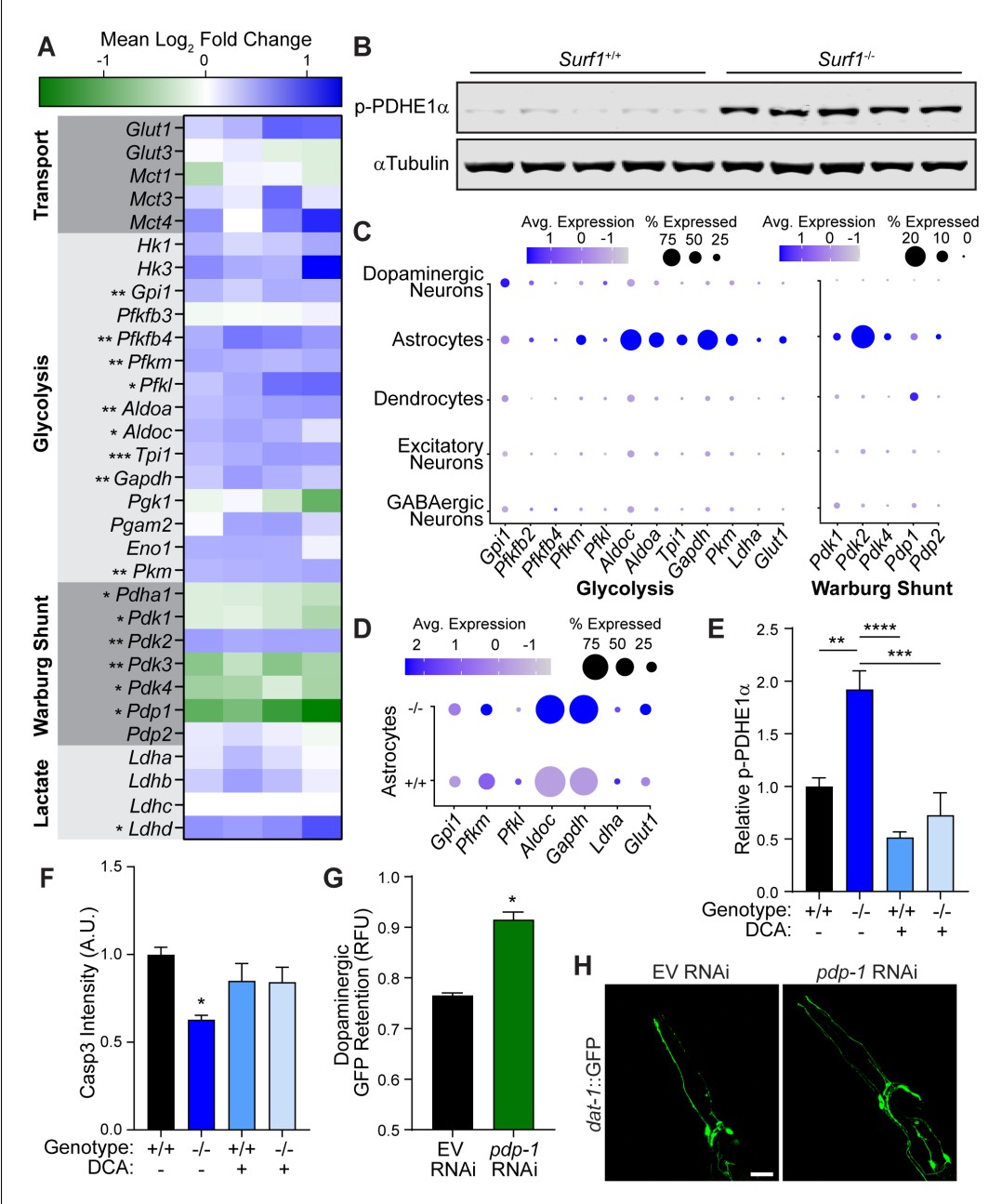

**Figure 5.** Impaired cytochrome C oxidase increases glycolysis in astrocytes through a Warburg-like effect. (A) Heat map of transcriptional differences in *Surf1*<sup>+/+</sup> versus *Surf1*<sup>-/-</sup> brains. n=4. Significance denoted next to gene. (B) Western blots of phosphorylated PDHE1α at Ser293 and αTubulin from *Surf1*<sup>+/+</sup> and *Surf1*<sup>-/-</sup> brains, 7 days post-injury. (C) Dot plot of single nuclei RNAseq of glycolytic and Warburg-related genes in neuronal cell subtypes from *Surf1*<sup>+/+</sup> and *Surf1*<sup>-/-</sup> brains. (D) Dot plot of single nuclei RNAseq of glycolytic genes in astrocytes from *Surf1*<sup>+/+</sup> and *Surf1*<sup>-/-</sup> brains. (E) Quantification of phosphorylated PDHE1α at Ser293 and αTubulin from *Surf1*<sup>+/+</sup> and *Surf1*<sup>-/-</sup> brains after NaCl and dichloroacetate (DCA) treatment. (F) Quantification of cleaved caspase-3 immunostaining in NaCl and DCA treated *Surf1*<sup>+/+</sup> (n=3) and *Surf1*<sup>-/-</sup> (n=4) brains 7 days post-injury. (G) Post-trauma retention of GFP fluorescence in dopaminergic neurons of *C. elegans* treated with *pdp-1* RNAi and measured by large-particle flow cytometry. n=1758 worms (EV RNAi), n=2307 worms (*pdp-1* RNAi) across two independent trials. (H) Representative micrographs of GFP dopaminergic neuronal morphology in injured *C. elegans* treated with *pdp-1* RNAi. Scale bar=20 μm. Data are mean ± SEM. *p ≤ 0.05, **p ≤ 0.01, ***p ≤ 0.001, and ****p ≤ 0.0001.

The online version of this article includes the following figure supplement(s) for figure 5:

**Figure supplement 1.** Warburg effect in concussive brain injury.

**Figure supplement 2.** Pyruvate dehydrogenase complex dynamics in blunt force trauma and concussive brain injury.

ROS levels observed in Surf1<sup>-/-</sup> brains (*Figure 4—figure supplement 1C*) reduce mitochondrial respiration in favor of glycolysis through *Hif1a* activation. Pharmacological inhibition of trauma-induced *Hif1a* activation has previously been shown to enhance necrotic lesion formation after brain trauma (*Umschweif et al., 2013*), suggesting a neuroprotective role for *Hif1a* activation in concussive injury. Consistent with previous studies in *C. elegans* (*Lee et al., 2010*), reducing cytochrome C oxidase activity through *cox-5b* RNAi increased *hif-1* activity as evidenced by induction of the established *hif-1* transcriptional target in *C. elegans*, *nhr-57* (*Figure 5—figure supplement 1L*). Supporting this conserved response, we observed HIF1α protein stabilization in Surf1<sup>-/-</sup> mouse embryonic fibroblasts (*Figure 5—figure supplement 1K*) as well as transcriptional upregulation of established glycolytic *Hif1a* targets in Surf1<sup>-/-</sup> brains, including critical enzymes in the Warburg shunt (*Figure 5A*).

Our data indicates that mild elevation of ROS caused by reduced ETC function, promotes a *Hif1a* mediated Warburg-like effect in the brain, which reduces the chance of overwhelming mitochondrial respiratory capacity under the strong energetic demand of TBI. To examine the necessity of this Warburg shift in our neuroprotective model, we administered the pharmacological Warburg inhibitor, dichloroacetate (DCA), and observed that *Surf1* mutants were no longer protected against trauma-induced neurodegeneration (*Figure 5E and F*; *Figure 5—figure supplement 1R*). Consistent with earlier reports of rodent brain injury (*Lazzarino et al., 2019*) and SURF1 mutants (*Figure 5A*), *pdp-1* expression was repressed in *C. elegans* after blunt force injury (*Figure 5—figure supplement 1I*). To further validate the Warburg shift as a neuroprotective mechanism, we employed genetic means in *C. elegans* and observed that *pdp-1* RNAi suppressed degeneration of dopaminergic neurons after trauma (*Figure 5G and H*). Albeit to a lesser degree, reduced expression of *pdp-1* exclusively in the nervous system was sufficient to promote survival of dopaminergic neurons after blunt force trauma (*Figure 5—figure supplement 2D*). Thus, modulating PDC activity through the key regulatory factors is sufficient to suppress trauma-induced neurodegeneration in both mice and worms.

## Discussion

The molecular details of TBI and its immediate and long-term consequences can be discovered using a translational animal model approach. To this end, we describe a comparative animal model to identify conserved molecular mechanisms operating in TBI. The worm *C. elegans* is amenable to high throughput genetic, biochemical, and behavioral studies, while the mouse is used to further uncover and validate conserved genetic targets and molecular mechanisms. These complementary models allowed us to characterize an evolutionary conserved, hypersensitive neuronal subtype to physical insult. Rather than being a stochastic mechanism of progressive tissue degeneration dictated by the type and site of injury, the physiological properties and regulatory mechanisms inherent to different neuronal subtypes likely determine their vulnerability to trauma. This is evidenced by sensitivity of dopaminergic neurons to biomechanical insult in both our randomized, high-frequency trauma model in worms and concussive injury in mice. Furthermore, these complementary models were used to demonstrate that genetic modification of a single gene can suppress degeneration of this neuronal subtype following concussive injury.

Pharmacological induction of Parkinsonian phenotypes, including death of dopaminergic neurons in the midbrain, requires highly oxidative drugs that irreversibly inhibit mitochondrial ETC complexes I and III (*Subramaniam and Chesselet, 2013*). Yet, our studies demonstrate that ETC complex IV insufficiency protects dopaminergic neurons from trauma-induced degeneration. We attribute these neuroprotective effects to a metabolic shift away from mitochondrial respiration, which minimizes or even abolishes transient oxidative stress in the neuron and suppresses dopaminergic degeneration. Although often thought of as unavoidable toxic byproducts of oxidative phosphorylation, ROS can have beneficial roles in the cell as a signaling molecule (*Guzy et al., 2005*). In our complementary models of blunt force injury, we believe that both toxic and protective mechanisms of ROS are operating, which historically has been difficult to discern.

Trauma-induced hyperglycolysis was reported decades ago yet its cellular resolution and pathophysiological role have remained unclear (*Bergsneider et al., 1997*; *Carpenter et al., 2015*). The astrocyte-neuron lactate shuttle is a crucial mechanism by which the brain maintains energetic homeostasis. Perturbations to this system have been reported in neurodegenerative disorders and contribute to disease progression (*Mason, 2017*). Our data suggests that preemptively repressing

mitochondrial respiration initiates a Warburg-like response in astrocytes as an attempt to restore homeostatic energetics. Thus, the injured brain likely induces hyperglycolysis in astrocytes as a protective mechanism to mitigate the oxidative damage resulting from over-activated mitochondrial respiration. Cerebral metabolic preconditioning presents a plausible prophylactic method for individuals at higher risk for TBI. Interestingly, neuroinflammation does not correlate with the rapid emergence of dopaminergic neurodegeneration in our study. However, neuroinflammation and other factors likely contribute to the later stages of this neurodegenerative condition (*McKee et al., 2015*). Therefore, longitudinal studies are required to further characterize disease progression, including neuroinflammation and proteotoxicity, after the initial loss of dopaminergic neurons in the midbrain.

## Materials and methods

### Mice

Adult mice of at least 8 weeks of age were used in this study. *Surf1*$^{+/+}$ and *Surf1*$^{-/-}$ (RRID:MGI: 3698949) were obtained from Massimo Zeviani (University of Cambridge). Littermates were obtained from *Surf1* heterozygous crossing and housed in groups of 5 regardless of phenotype with food and water ad libitum. All mouse studies were approved by the University of Texas Southwestern Medical Center Institutional Animal Care and Use Committee (Protocol No. 2016–101750) and performed in accordance with institutional and federal guidelines.

### Closed-head traumatic brain injury

The closed-head traumatic brain injury device consisted of an upright, railed-guided weight drop composed of an aluminum body and equipped with a speed monitor device to measure the velocity of the weight drop when it lands on a brass impactor with a slightly concave nylon tip. The modular weight system consisted of a 50 ml polypropylene conical tube filled with lead buckshot, which weighted 220 g when filled. Before injury, mice were anesthetized with vaporized isoflurane/oxygen in a chamber connected to an inhalation anesthesia system (Cat. No. 901810; VetEquip). Vaporizer settings were set to 3% USP grade isoflurane and 2.5 lpm USP grade oxygen. After being anesthetized, mice were placed on a thick memory foam cushion resting on a height-adjustable platform. Once the desired site of impact was localized on the head of mice, the impactor tip was placed directly in contact with that site and the weight was dropped by activating an electronic trigger. After mice were injured, they were placed on their backs to record their righting reflex recovery time and allowed to recover. After recovery, mice were placed back in their cage with food and water ad libitum. Control mice received only anesthesia and were not subjected to impact, but their righting reflex recovery time was also recorded in the same manner as the injured mice.

### Finite element analysis

Biomechanical responses of the mouse brain under impact were analyzed using the commercial finite element (FE) code ABAQUS (Dessault Systemes Simulia Corp., Providence RI). More specifically, a 2D geometric model of the sagittal section of the mouse brain tissue was first generated based on a partition scheme defined previously (*MacManus et al., 2016*) and the 2D mouse brain image obtained from the experiment. This model also contains a layer of skull of thickness of 1.5 mm. The 2D model was meshed with 5067 4-noded quadrilateral elements, which was then converted to 3D by extruding the 2D meshed plane orthogonally that gives a layer thickness of 16 mm. The 3D geometric model of the impactor was constructed based on the measured dimensions shown in (*Figure 2—figure supplement 1A*), which is meshed with 40,320 8-noded brick elements (C3D8R). The final 3D FE model of the brain was modeled with 2,546,048 8-noded brick elements (C3D8R) (*Figure 2—figure supplement 1B*). The brain tissue was partitioned into six regions (*MacManus et al., 2016*), and 6 sets of viscoelastic parameters from P56 mice (*MacManus et al., 2017*) were applied to model the different viscoelastic properties of corresponding regions. Viscoelastic parameters from adult rat (*Finan et al., 2012*) had to be used to define the hippocampus due to lack of characterization in the mouse. Values of these parameters are listed in *Tables 1* and *2*.

To start the simulation, the impactor was placed at 0.9 mm above the skull with initial velocity of 4.54 m/s. Dynamic responses of the mouse brain were computed with the standard explicit time

**Table 2.** Material parameters for the impactor.

| Component | Density | Young's Modulus | Poisson's Ratio |
|---|---|---|---|
| Brass (impactor body, head) | 8480 kg/m³ | 100 GPa | 0.34 |
| Nylon (impactor nut) | 1120 kg/m³ | 2.3 GPa | 0.4 |

integration scheme with a time step of 2 ms and total steps of 200. Interaction between the impactor and skull was modeled by the frictionless and surface-to-surface contact algorithm. Contact between the skull and brain tissue was treated with tie constraint. Five points in cortex, thalamus, substantia nigra, cerebellum, and brainstem were selected to monitor the time history of the principal strain and effective stress during impact (*Figure 2—figure supplement 1B*). Points for the cortex, thalamus, and substantia nigra were respectively located 2.4 mm, 4.1 mm, and 8.2 mm in vertical distance to the point of impact. Points for the cerebellum and brainstem were located 4.7 mm posterior to the three previous points, 4.1 mm and 8.2 mm in vertical distance to the point of impact. The impactor contained brass and nylon parts, both of which were modeled as isotropic linear elastic material. Material properties of the two are provided in *Table 2*. The skull is also modeled as isotropic linear elastic material with Young's Modulus of 1 GPa and Poisson's ratio of 0.33 (*Unnikrishnan et al., 2019*). Mass density skull is 1710 kg/m³ (*Hua et al., 2015*).

Mass density of the brain tissue is 1040 kg/m³ and it is modeled as a viscoelastic material (*Hua et al., 2015*) in which stress is evaluated through the introduction of the relaxation function $G(t)$:

$$\tau(t) = \int_0^t G(t-s)\dot{\gamma}(s)ds \tag{1}$$

in which $\gamma$ is the shear strain and superimposed dot representing the time derivative. The relaxation function $G(t)$ takes the form of the Prony series, given as

$$G(t) = G_\infty + \sum_{i=1}^N G_i e^{-t/\tau_i} \tag{2}$$

in which $t$ is time, $G_\infty$ is the long-term relaxation modulus, $\tau_i$ is the characteristic relaxation time and $G_i$ is the corresponding relaxation modulus. The number of the terms $N$ included in this series depends on fitting of the model predictions to the experiments (*Finan et al., 2012*).

## Immunohistochemistry

Mice were perfused with cold 1x PBS and 4% paraformaldehyde. Brains were harvested and sectioned to a thickness of 40–50 µm with the VF-310-0Z Compresstome Vibrating Microtome (Precisionary Instruments, Greenville, NC). All staining was performed on free floating brain sections on a 24-well plate. Incubation in primary antibodies was performed in blocking solution (10% normal donkey serum) and 2% triton at 4°C for 48 hr. Primary antibodies used were: rabbit anti-Iba1 (Cat. No. 019–19741; Wako Chemicals; RRID:AB_839504) at 1:200, rat anti-CD68 (Cat. No. MCA1957; AbD Serotec; RRID:AB_322219) at 1:250, chicken anti-tyrosine hydroxylase (Cat. No. TYH; Aves Labs Inc; RRID:AB_10013440) at 1:1000; rabbit anti-cleaved caspase-3 (Cat. No. 9661S; Cell Signaling Technologies; RRID:AB_2341188) at 1:400. Species-specific secondary antibodies (Jackson ImmunoResearch/Invitrogen) were used at a concentration of 1:250 at 4°C for 24 hr. Nissl staining (Neurotrace 530/615; Cat. No. N21482; Invitrogen) was performed according to manufacturer's instructions.

## Metabolic phenotyping

The metabolic phenotyping experiments were run by the Metabolic Phenotyping Core of UTSW Medical Center in metabolic cages using a TSE Systems, Inc (Chesterfield, MO) indirect calorimetry system. Briefly, the day following a single concussive brain injury, mice were singly housed in shoebox-sized cages with wood chip bedding to acclimate for 5 days, followed by 5 days of experimental recording. $O_2$ consumption and $CO_2$ production were measured to determine energy expenditure and respiratory exchange ratio. Overall physical activity was determined through x, y, and z beam

breaks. Body heat production, food intake, and water consumption were also measured. Data was normalized according to body weight.

## Metabolome extraction and profiling

Metabolomics profiling was performed by the Metabolomics Facility at the Children's Medical Center Research Institute at UT Southwestern. Briefly, after concussive brain injury, brains were extracted, midbrains were dissected and flash frozen in liquid nitrogen. Tissue was then homogenized in a Precellys homogenizer at 5000 rpm for 30 s (2x) with a 20 s pause at 4°C. The metabolome was then extracted with an 80% ice cold solution of methanol and $ddH_2O$. The metabolite containing supernatant was transferred into a clean centrifuge tube and desiccated for 12 hr. Metabolome profiling was done via LC-MS, normalized to total ion current and SIMCA analyzed.

## Protein extraction and western blotting

Tissue or cells were homogenized in a Precellys homogenizer at 4 °C with ceramic beads at 5,000 rpm for 30 s (2x) with a 20 s pause using RIPA lysis buffer (150 mM NaCl, 5 mM EDTA, 50 mM Tris, 1% NP-40, 0.5% sodium deoxycholate, 1% sodium dodecyl sulfate (SDS), final pH 8) supplemented with cOmplete EDTA-free mini-protease inhibitor cocktail (Cat. No. 11836170001; Roche). Extracts were created in the presence of the 1% SDS detergent to ensure protein linearization and inactivation of phosphatases (*Stinson, 1984*). Protein samples were centrifuged for 10 m at 10,000 rcf at 4 °C and the supernatant was used for protein quantification via Pierce BCA protein assay (Cat. No. 23225; Thermo Fisher Scientific). All sample concentrations were standardized and diluted in sample buffer. Samples were boiled at 90°C for 10 m, resolved by SDS-PAGE, transferred to nitrocellulose membranes and subject to western blot analysis. SDS-PAGE gels (10%) were prepared the night before and stored at 4°C. The EZ-Run pre-stained protein ladder (Cat. No. BP3603500; Thermo Fisher Scientific) was loaded and electrophoresis was performed at 100 V until the 40 kDa protein standard reached the bottom of the gel. All antibodies were prepared in 5% BSA/PBST. Mouse anti-αTubulin (Cat. No. T6074; Sigma; RRID:AB_477582) was used at 1:10000, rabbit anti-phospho-PDHEα1 (Cat. No. 31866S; Cell Signaling Technologies; RRID:AB_2799014) was used at 1:1000, and rabbit anti-HIF1α (Cat. No. 3716S; Cell Signaling Technologies; RRID:AB_2116962) was used at 1:1000. Western blots were quantified using Image Studio software (LI-COR Biosciences, Lincoln, NE). Quantified bands of interest were standardized based on band signal intensity of αTubulin.

## ATP levels measurement

ATP levels were measured with the CellTiter-Glo luminescence assay (Cat. No. G7570; Promega) according to manufacturer's instructions. Brains were harvested, enzymatically dissociated with papain, and passed through a cell strainer (45 μm). Myelin was removed with the Debris Removal Solution (Cat. No. 130-109-398; Miltenyi Biotec). Freshly dissociated cells (100,000 cells) were resuspended in 100 μl of 1x PBS and transferred to a 96-well plate. A total of 100 μl of CellTiter-Glo reagent was added to make a final volume of 200 μl. The 96-well plate was placed on an orbital shaker for 2 m to induce cell lysis. After a 10 m incubation at room temperature, the plate was read for luminescence (0.25 s per well) in a CLARIOstar Plus microplate reader (BMG LABTECH, Ortenberg, Germany).

## Reactive oxygen species detection

Mitochondrial superoxide in mouse brain single cell suspension was measured with the MitoSOX Red superoxide indicator (Cat. No. M36008; Invitrogen) according to manufacturer's instructions. Briefly, brains were harvested, enzymatically dissociated with papain, and passed through a cell strainer (45 μm). Myelin was removed with the Debris Removal Solution (Cat. No. 130-109-398; Miltenyi Biotec). Freshly dissociated cells were then incubated in a 5 μM MitoSOX solution in a 96-well plate for 10 m at 37°C. Each well was measured for fluorescence (Ex/Em: 510/580 nm). Reactive oxygen species were measured in *C. elegans* with the Amplex Red Hydrogen Peroxide/Peroxidase Assay Kit (Cat. No. A22188; Invitrogen). Worms were grown to Day one adults, transferred to a 2 ml centrifuge tube containing 500 μl of M9, and submitted to traumatic injury. Immediately after traumatic injury, equal number of worms (50 μl total M9 volume) were transferred to a 96-well plate (three technical repeats) and 50 μl of 100 μM Amplex Red reagent/0.2 U/ml horseradish peroxidase

was added to a final concentration of 50 µM Amplex Red reagent/0.1 U/ml horseradish peroxidase. Each well was measured for fluorescence (Ex/Em: 530–560/~590 nm) every 92 s for 1.5 hr on a CLAR-IOstar Plus microplate reader (BMG LABTECH).

### *C. elegans* strains

*C. elegans* were maintained as previously described (*Brenner, 1974*). Worm strains were expanded on nematode growth media (NGM) plates supplemented with *Escherichia coli* OP50. Worm strains that were used for experimental purposes were grown on NGM plates supplemented with HT115 *E. coli* at 20°C (see RNAi Administration). For temperature sensitive strains, worms were grown at 25°C. The following *C. elegans* strains were generated in our laboratory and used in this study: PMD13 (*egIs1[dat-1p::GFP]; rrf-3(b26); fem-1(hc17)*), PMD74 (*unc-119p::TOM20::KillerRed; egIs1[dat-1p:: GFP]; rrf-3(b26); fem-1(hc17)*), PMD63 (*egIs1[dat-1p::GFP]; rrf-3(b26); fem-1(hc17); uIs69 [pCFJ90 (myo-2p::mCherry) + unc-119p::sid-1]; sid-1(pk3321)*).

### *C. elegans* trauma

Trauma was administered to *C. elegans* as previously described (*Egge et al., 2021*). Briefly, age-synchronized worms were grown on empty vector (EV) or the respective RNAi until day 1 of adulthood. Temperature-restrictive strains were grown at 25°C to avoid generation of progeny. Worms were rinsed off growth plates in liquid M9 buffer (M9) and pelleted by centrifugation at 1000 x g for 30 s. 100 µl worm pellets were then transferred to 2 ml Precellys tubes (Cat. No. 02-682-556; Thermo Fisher Scientific) in a total volume of 500 µl of M9. Worms were subject to high-frequency, multidirectional agitation in a Precellys Evolution homogenizer (Cat. No. P000062-PEVO0-A.0; Bertin Instruments) for 16 s at 8600 rpm. Worms were pelleted at 1000 x g for 30 s and transferred to recovery plates containing the respective RNAi or EV on which they were cultured prior to trauma. Control, non-injured worms were suspended in M9 for comparable lengths of time and then transferred to recovery plates without having received trauma.

## Confocal imaging

Worms were paralyzed with 1 mM levamisole and mounted on microscope slides with M9 buffer. Brain sections were mounted in microscope slides with Fluoromount-G with DAPI (Cat. No. 00-4959-52; Invitrogen). Confocal images were collected with a Leica SP8 confocal microscope equipped with one photomultiplier tube, two super-sensitive hybrid HyD detectors, and highly stable lasers (UV/405 nm DMOD compact, 488 nm, 552 nm, 638 nm). The following Leica PL APO CS2 objectives were used to collect images: air 10x/0.40 NA, oil 40x/1.30 NA, oil 63x/1.40 NA. Super-resolution imaging was achieved with the Leica LIGHTING module built into the Leica LAS X software. Brain parenchyma images were collected as Z-stacks with 1 µm steps as follows: midbrain micrographs 20 µm/10x objective/5 tile Z-stacks, cortex micrographs 10 µm/40x objective/5 tile Z-stacks, hippocampus micrographs 20 µm/10x objective/8 tile Z-stacks, thalamus micrographs 5 µm/40x objective/2 tile Z-stacks. Number of tiles to be collected was determined by completely covering the brain structure of interest (i.e. midbrain 1x5 tiles, hippocampus 2x4 tiles). Tiled images were stitched with the Leica LAS X software. Cell culture micrographs were collected from cells grown on glass coverslips with a total of 10 fields collected with a 40x objective for image processing and analysis. Brain parenchyma images are displayed as maximal intensity projections of Z-stacks. Cell culture images are displayed as single plane micrographs. All confocal images were processed and analyzed with ImageJ (NIH, Bethesda, MD; RRID:SCR_003070).

## Motor analysis (Rotarod)

Motor deficits in mice were measured by Rotarod (Cat. No. 76–0770; Harvard Apparatus) with a rod diameter of 3 cm and a rod height of 20 cm. Animals were not trained on a rotating rod but rather exposed to a static Rotarod machine 2 days before testing for 2 m, three times daily. For testing, animals were placed on the rod with accelerating speed (0–24 rpm in 120 s) for all experiments. Latency to fall was recorded electronically in seconds by the apparatus and values were averaged per group for reporting. The clock was stopped if an animal held to the rod on two consecutive rotations and if the animal failed to fall after 120 s. Animals were returned to their cage after each trial.

Ambulation was measured via laser beam breaks for 5 days in mice housed individually in cages equipped with laser beams and other probes meant to determine changes in metabolism.

## RNA extraction

Four biological repeats of age-matched adult $Surf1^{+/+}$ and $Surf1^{-/-}$ brains were collected for each of the following conditions: $Surf1^{+/+}$ uninjured, $Surf1^{+/+}$ injured, $Surf1^{-/-}$ uninjured, $Surf1^{-/-}$ injured at time-points 2 hr post-injury and 7 days post-injury. Mouse brains were harvested in TRIzol (Cat. No. 15596018; Thermo Fisher Scientific), flash-frozen in liquid nitrogen, and stored at −80˚C. Brains were triturated with syringes and freeze-thawed three times, followed by a chloroform/isopropanol extraction process. The RNA pellets were washed twice with ethanol, air-dried, and reconstituted in 20 µl ddH$_2$O. RNA quality (260/280, 260/230 ratio) and concentration was determined using the DS-11 FX+ spectrophotometer (DeNovix, Inc, Wilmington, DE).

## Illumina sequencing RNAseq analysis

Quality control, mRNA purification, and paired-end 150 bp Illumina sequencing were performed by Novogene (Sacramento, CA). mRNA was enriched using oligo(dT) beads, randomly fragmented in fragmentation buffer, and reverse transcribed to cDNA using random hexamers. Following first-strand synthesis, Illumina sysnthesis buffer was added with dNTPs, RNase H, and *E. coli* polymerase I to synthesize the second strand by nick-translation. The cDNA library was purified, underwent terminal repair, A-tailing, and ligation of adapters before PCR enrichment. The cDNA library concentration was quantified with a Qubit 2.0 fluorometer (ThermoFisher Scientific) and sized with an Agilent 2100 Bioanalyzer (Agilent, Santa Clara, CA). RNAseq statistical analysis was performed using CLC software (version 9.0, CLC Bio, Aarhus, Denmark). Data is presented as reads per kilobase million (RPKM) with values normalized to control levels and made relative to 1.

## Single nuclei isolation, purification, and RNA sequencing

Single nuclei isolation methods were adapted from 10x Genomics protocols CG000212 Rev B and CG000124 Rev D. $Surf1^{+/+}$ and $Surf1^{-/-}$ brains were harvested under cold sterile 1x Dulbecco's PBS (without Ca$^{2+}$ and Mg$^{2+}$). Immediately after harvesting, the cortex, olfactory bulbs and cerebellum were removed on an ice block and discarded. Brains were then triturated and lysed using the Nuclei PURE Prep Isolation Kit (Cat. No. NUC-201, Sigma). Briefly, brains were homogenized under cold Nuclei PURE Lysis Buffer and mechanically triturated with sterile pipettes and pipette tips of decreasing sizes until creating a uniform suspension. Suspensions were then filtered through 70 µm, 40 µm, and 30 µm cell strainers to remove cell debris. Myelin removal proceeded using the Debris Removal Solution (Cat. No. 130-109-398, Miltenyi) according to manufacturer's instructions. Single nuclei were isolated and purified via density gradient centrifugation with the Nuclei PURE sucrose cushion solution (Cat. No. NUC-201, Sigma) at a centrifugation speed of 13,000 rcf. Nuclei was resuspended in 1x Dulbecco's PBS supplemented with 1% bovine serum albumin and 20 U/µl RNase inhibitor (Cat. No. AM2694, Invitrogen). Single nuclei suspensions were submitted to the Next Generation Sequencing Core at UT Southwestern Medical Center for sequencing and library preparation. Briefly, nuclei were loaded with Single Cell 3' Gel Beads into a Next GEM Chip G and run on the Chromium Controller. GEM emulsions were incubated and then broken. Silane magnetic beads were used to clean up GEM reaction mixture. Read one primer sequence was added during incubation and full-length, barcoded cDNA was then amplified by PCR after cleanup. Sample size was checked on the Agilent Tapestation 4200 using the DNAHS 5000 tape and concentration was determined by the Qubit 4.0 Fluorimeter (ThermoFisher) using the DNA HS assay. Samples were enzymatically fragmented and underwent size selection before proceeding to library construction. During library preparation, Read two primer sequence, sample index, and both Illumina adapter sequences were added. Subsequently, samples were cleaned up using Ampure XP beads and post library preparation quality control was performed using the DNA 1000 tape on the Agilent Tapestation 4200. Final concentration was ascertained using the Qubit DNA HS assay. Samples were loaded at 1.6 pM and run on the Illumina NextSeq500 High Output Flowcell using V2.5 chemistry. Run configuration was 28x98x8.

## Single nuclei RNA-seq data analysis

Raw gene counts were obtained by aligning the FASTQ files to *Mus musculus* mm10 as reference genome using CellRanger Software (v5.0.0) and then analyzed using Seurat-3.2.0 (*Stuart et al., 2019*). Genes detected in <0.2% of the nuclei and from mitochondrial and sex chromosomes (X and Y) were filtered out. Nuclei that had >12,000 UMI, <300 genes, and/or >12% mitochondrial content were further excluded. After quality control, 5140 nuclei (primary dataset) were further analyzed for their gene expression profiles. Post filtering, the expression values were log-normalized, scaled with a factor of 10,000 and regressed to covariates (percent mitochondrial content and number of genes per nuclei), with Seurat's SCTransform method and integrated. The nuclei were then assessed by Principal Component Analysis (PCA) dimensionality reduction, followed by shared nearest neighbor (SNN) modularity optimization (Louvain algorithm) based clustering algorithm to identify the clusters. The clusters were visualized using the Uniform Manifold Approximation and Projection (UMAP) dimensional reduction technique. Cell types were assigned to clusters based on enrichment of marker genes from top expressed genes as follows: Excitatory neurons (*Rbfox3*), GABAergic neurons (*Gad1,Gad2*), Dendrocytes (*Cacng4*), Astrocytes (*Gaj1, Aqp4*), and Dopaminergic neurons (*Slc6a3, Slc18a2, Ddc*). Astrocyte and neuronal clusters were subclustered (secondary dataset) and then exclusively studied using the same approach as described above. Pairwise differential gene expression analysis tests were performed with Wilcox test within Seurat. The expression of specific genes within clusters were visualized using the DotPlot function in Seurat. The average expression of these genes in the dot plot was calculated by using the normalized gene counts, natural-log transforming them using log1p, and then scaling them using a z-transformation.

## RNAi administration

Worm strains were grown on HT115 *E. coli* harboring RNAi constructs from either the Ahringer or Vidal RNAi libraries (*Rual et al., 2004*). The L4440 empty vector (EV) RNAi construct was used for control treatments. RNAi strains were grown in small cultures before inoculating larger cultures in Terrific broth (TB) and grown for 15 hr on an orbital shaker at 37°C. After 15 hr, cultures were treated with 1 mM IPTG and incubated for an additional 4 hr at 37°C to induce expression. Cultures were then centrifuged at 4000 x g and bacterial pellets were re-suspended to specific concentrations in TB before being spread on 100 mm NGM plates containing a final concentration of 1 mM IPTG and 0.1 mg/ml carbenicillin. Optical density (OD600) of RNAi-producing bacterial cultures was used to standardized and equalize cell number for multiple RNAi construct combinations. For titrated single RNAi construct treatments, induced HT115 bacteria expressing a single RNAi construct was diluted with L4440 empty vector HT115 bacteria.

## Large-particle flow cytometry

Large-particle flow cytometry was performed as previously described (*Egge et al., 2019*). Briefly, flow cytometry of *C. elegans* was performed on a COPAS FP-250 flow cytometer (Union Biometrica, Holliston, MA) and an automated sample introduction system LP Sampler (Union Biometrica) fitted with 96-well plates. M9 buffer was utilized as the sample solution for worm flow. The COPAS GP sheath reagent (PN: 200-5070-100, Union Biometrica) was used as sheath solution. Prior to every run, laser power and flow rates are calibrated by use of GP control particles (Cat. No. 310-5071-001; COPAS Biosorter, Union Biometrica) as recommended by the manufacturer. From experiment to experiment, we maintain a consistent PTM laser power for each respective fluorophore. Flow data was collected in FlowPilot software (Union Biometrica). Details regarding analysis and normalization of data obtained from large-particle cytometry are found under the 'Statistical analysis' methods section.

## Cell culture

HEK293t cells were obtained from ATCC (Cat. No. CRL-3216; RRID:CVCL_0063) and cultured in DMEM/F-12 (Cat. No. 11320033; Gibco) supplemented with 10% Fetal Bovine Serum and 1x Penicillin-Streptomycin (Cat. No. P4333; Sigma). Mouse embryonic fibroblasts (MEF) were obtained via timed mating by detection of a vaginal plug was used to obtain embryonic day 13.5 embryos. Embryos were dissected in cold 1x PBS to remove the head and all inner organs, leaving only the carcass. After finely mincing the carcass in cold 1x PBS, the pieces were allowed to settle, and PBS

was aspirated. The tissue was then enzymatically dissociated with 1 ml of 0.25% trypsin/EDTA at 37°C for 5 m. The tissue was then mechanically dissociated using a 1 ml pipette tip until homogeneous and passed through a nylon cell strainer (45 µm). Cells were cultured in DMEM-Hi glucose (Sigma-Aldrich; 4.5 g/L glucose, supplemented with L-glutamine and sodium pyruvate), 10% fetal bovine serum (heat inactivated), and supplemented with penicillin streptavidin. All cultures were confirmed to be negative for mycoplasma with the MycoAlert Mycoplasma Detection Kit (Cat. No. LT07-218; Lonza).

## Oxygen consumption rate

$Surf1^{+/+}$ and $Surf1^{-/-}$ mouse embryonic fibroblasts were plated on a clear bottom 96-well plate (100,000 cells/well). Each well was treated with 100 ng/ml of phorbol myristate acetate (PMA) for 1 hr. The PMA containing medium was then replaced with fresh medium and the basal oxygen consumption rate was measured with the Cayman Chemical Oxygen Consumption Rate Assay Kit (Cat. No. 600800) according to manufacturer's instructions.

## Media acidification assay

Mouse embryonic fibroblasts (MEF) were cultured in a 96-well plate. Each well contained equal MEF numbers cultured in DMEM/F-12 (Cat. No. 11320033; Gibco) supplemented with 10% Fetal Bovine Serum. DMEM/F-12 used contains phenol red as a pH indicator with pH ranges of 6.8 (yellow) to 8 (red). Cells were cultured for 5 d in a cell culture incubator, at 37°C with 5% $CO_2$. After 5 days, the Optical Density was immediately measured in a CLARIOstar Plus spectrometer (BMG LABTECH) by performing a spectral scan between 220 nm and 1000 nm with a 1 nm resolution. Data was normalized to $1 \times 10^5$ cells for analysis.

## LDH activity assay

The maximum LDH activity was measured with the Pierce LDH Cytotoxicity Assay Kit (Cat. No. 88953; Thermo Fisher Scientific) according to manufacturer instructions. Briefly, 20,000 MEF were cultured overnight in a flat bottom, 96-well plate at 37 °C, 5% $CO_2$. Cells were lysed with lysis buffer provided in the kit and 50 µl of supernatant was transferred to a fresh 96-well flat bottom plate. The kit's proprietary reaction solution was added to the supernatant and incubated at room temperature for 30 m. After stopping the reaction, absorbance was measured at 490 nm and 680 nm in a CLARIOstar Plus microplate reader (BMG LABTECH). LDH activity was determined by subtracting the 680 nm from the 490 nm absorbance values.

## Statistical analysis

All statistical analyses were performed using Prism eight software (GraphPad, San Diego, CA) unless noted. Student's t-test was used to compare means between two normal populations. Mann-Whitney U test was used to compare differences in the dependent variable between two groups. Post-hoc analysis performed after ANOVA included Dunnett's multiple comparison (to compare means from several experimental groups against a control group mean). Tukey's multiple comparisons test (to compare all possible pairs of means).

Statistical analysis of large-particle flow cytometry data was performed in Prism and Excel (*Egge et al., 2021*). To evaluate large worm numbers across biological repeats, error propagation was performed according to the general formula:

$$\delta R = \sqrt{\left(\frac{\partial R}{\partial X} * \delta X\right)^2 + \left(\frac{\partial R}{\partial Y} * \delta Y\right)^2 + \ldots}$$

where $\delta R$, the total error within each independent repeat, is a function of each independent variable ($X, Y \ldots$). For the particular case of error propagation for the Dopaminergic GFP Index within each of multiple biological repeats, error is further added in quadrature and the function becomes:

$$\delta R = \sqrt{\sum_X \left(\frac{\delta X}{1 - EV}\right)^2}$$

where $\delta X$ represents the error and $X$ contains all biological repeats. *EV* is a constant determined by

the relative loss of fluorescence observed in injured worms grown on EV bacteria during that experiment. Lastly, the total error within each repeat was added to the error between repeats in quadrature.

## Acknowledgements

We thank Massimo Zeviani, PhD at the University of Padua for providing *Surf1* mutant mice. We thank Lauren Zacharias, MS and Ralph DeBerardinis, MD, PhD of the Children's Medical Center Metabolomics Facility at UT Southwestern Medical Center (UTSW) for targeted metabolomic profiling and advice on data analysis. We thank Syann Lee, PhD at the Metabolic Phenotyping Core at UTSW for overseeing mouse metabolic cage experiments. We thank Matt Seiber, PhD, William Dauer, MD, Munro Cullum, PhD at UTSW, and Michael Douglas, PhD for critical feedback. Funding: Our funding sources are from the Clayton Foundation for Research, Welch foundation (I-2061–20210327 to PMD), the American Federation of Aging Research, the Glenn Center for Aging, the NIH (R00AG042495 and R01AG061338 to PMD), and the Cancer Prevention Research Institute of Texas (CPRIT) (RR150089 to PMD).

## Additional information

### Competing interests

Genevieve Konopka: Reviewing Editor, eLife. The other authors declare that no competing interests exist.

### Funding

| Funder | Grant reference number | Author |
| --- | --- | --- |
| Welch Foundation | I-2061-20210327 | Peter M Douglas |
| National Institutes of Health | R01AG061338 | Peter M Douglas |
| Cancer Prevention and Research Institute of Texas | RR150089 | Peter M Douglas |
| Clayton Foundation for Research | | Peter M Douglas |
| National Institutes of Health | R00AG042495 | Peter M Douglas |

The funders had no role in study design, data collection and interpretation, or the decision to submit the work for publication.

### Author contributions

Rene Solano Fonseca, Conceptualization, Data curation, Formal analysis, Investigation, Methodology, Writing - original draft, Writing - review and editing; Patrick Metang, Nathan Egge, Yingjian Liu, Karthigayini Sivaprakasam, Formal analysis, Investigation; Kielen R Zuurbier, Formal analysis, investigation.; Shawn Shirazi, Resources, Methodology; Ashleigh Chuah, Investigation; Sonja LB Arneaud, Investigation, Writing - original draft, Project administration, Writing - review and editing; Genevieve Konopka, Formal analysis, Methodology; Dong Qian, Conceptualization, Formal analysis, Investigation, Methodology, Writing - review and editing; Peter M Douglas, Conceptualization, Data curation, Formal analysis, Supervision, Funding acquisition, Writing - original draft, Writing - review and editing

### Author ORCIDs

Rene Solano Fonseca https://orcid.org/0000-0002-5927-5936
Sonja LB Arneaud http://orcid.org/0000-0002-1123-3876
Genevieve Konopka http://orcid.org/0000-0002-3363-7302
Peter M Douglas https://orcid.org/0000-0002-0734-1049

## Ethics

Animal experimentation: All mouse studies were approved by the UT Southwestern Medical Center Institutional Animal Care and Use Committee (IACUC) protocols (#2016-101750) and performed in accordance with institutional and federal guidelines.

## Decision letter and Author response

Decision letter https://doi.org/10.7554/eLife.69438.sa1
Author response https://doi.org/10.7554/eLife.69438.sa2

# Additional files

## Supplementary files

- Source data 1. Uncropped western blot images.

- Transparent reporting form

## Data availability

All datasets are submitted to GEO and will be made available to the public upon publication of the article.

The following datasets were generated:

| Author(s) | Year | Dataset title | Dataset URL | Database and Identifier |
|---|---|---|---|---|
| Douglas PM, Fonseca RS | 2021 | Glycolytic preconditioning in astrocytes mitigates trauma-induced neurodegeneration | https://www.ncbi.nlm.nih.gov/geo/query/acc.cgi?acc=GSE173431 | NCBI Gene Expression Omnibus, GSE173431 |
| Douglas P, Konopka G, Sivaprakasam K, Fonseca RS | 2021 | Glycolytic preconditioning in astrocytes mitigates trauma-induced neurodegeneration | https://www.ncbi.nlm.nih.gov/geo/query/acc.cgi?acc=GSE179905 | NCBI Gene Expression Omnibus, GSE179905 |

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
