## [Decision Letter]

**Acceptance summary:**

The results presented in this work will be of broad interest to readers in the field of neurodegeneration. The authors detail a neuronal subtype specific response to blunt force trauma in two different animal models. They also uncover a conserved shift in cellular metabolism that plays a protective role against neuronal death following injury.

**Decision letter after peer review:**

Thank you for submitting your article "Glycolytic preconditioning in astrocytes mitigates trauma-induced neurodegeneration" for consideration by *eLife*. Your article has been reviewed by 3 peer reviewers, and the evaluation has been overseen by a Reviewing Editor and Matt Kaeberlein as the Senior Editor. The reviewers have opted to remain anonymous.

Summary:

The reviewers and I are in agreement that the subject of the manuscript is of high interest and that the data largely support the assertions, but some assertions are less well supported and methodological details and some controls are insufficient. We believe this is a worthwhile manuscript for publication in *eLife*, but should be revised with some additional data and controls. In terms of additional experiments/data, we would ask that you focus on those that are crucial to the claims made within the text. These include the essential revisions listed below. While many of these are textual changes, some of these will require new data including data the first 4 points listed below. It was also pointed out that the n was very low (n=3) for many experiments, and while it is not feasible to ask to increase the n across all experiments, please very clearly note in the text where the low n could affect interpretation or reproducibility of the results. All of the reviewer comments are also listed below, and we would also strongly recommend making changes for the additional clarity and typographical changes recommended in the reviews that would improve the manuscript.

Essential revisions:

Additional experiments:

1. Caspase 3 staining following injury should be done for thalamus and cortex of injured mice to show that the neurodegeneration is midbrain dopaminergic neuron specific, rather than just the hippocampus, which is farther than the midbrain from the site of impact.

2. Clarification or modification/retesting of the worm cox5b RNAi protocol: As written, worms are injured and then placed into RNAi lawns to suppress de novo translation of cox5b to the effect of mitigating ROS production. However, this does not make sense, since ROS production should be able to be generated with existing mitochondrial protein complexes and not necessarily require new protein. This would suggest the vast literature of "tired old mitochondrial proteins are the cause of ROS" is collectively wrong. Resolve this apparent conflict. Possible methods may use cycloheximide to halt general translation or use of tet on/off systems controlling target protein expression.

3. Restaining and/or reimaging figure 1C (see below).

4. Panel 5B: Please run all samples on a single western blot and show results as a single boxed area. When western blots are presented such as this it is difficult to determine an effect of "injury".

Textual revisions: I strongly recommend reading reviewer 3's thorough review and verifying/modifying all of the typographical / nomenclature/ clarity changes requested.

1. Line 46: Unclear statement. Clarify

2. Line 113-115: please temper statement

3. Line 139: modify for clarity.

4. Line 159 to 161: modify for clarity.

5. Line 219 to 221: temper statement.

6. Line 82: Change "degeneration" to "injury".

7. Line 175/190: Was a correlation statistically evaluated? If so, please report results. If not, change wording. Perhaps use "corresponded".

8. Line 203: Replace "activation" with something more precise such as "upregulation" or "increased transcription". The word activation implies signaling or physiological response.

9. Line 212: Change "spatial" to "cellular".

10. Line 219 to 221: soften claim.

Figure/Table Revisions:

Please provide timepoints for all relevant figures where not listed. Also please carefully label control and TBI mice in each panel where relevant, and provide n where not clear.

Table 1. Better describe and clarify if parameters are for mouse brain or rat brain. If they are for rat, then provide the correct parameters for mouse brain. If the title is in error, then correct the title.

Figure 1. Panel C: Requires a reference map or annotations directly on the image to determine the precise anatomical location(s) pictured in the micrographs. If this is the midbrain Substantia nigra, please label SNp, SNr, VTA, etc. If this is not precisely in that area, additional tissue staining may be required to identify cell types. TH (tyrosine hydroxylase) is a marker of catecholaminergic cells and not specifically a marker of dopaminergic cells. Staining with TH will therefore include dopaminergic and noradrenergic cells which vary in their distribution throughout the midbrain to pons. If needed use dopamine β-hydroxylase (DBH) in addition to TH to differentiate dopaminergic cells from noradrenergic cells.

Panel E: It is unusual for uninjured mice to fall off the rotorod so quickly, especially when they were previously trained on the task two days prior. Provide data for all tested speeds if available.

Both Table 1 and Figure S2 indicate the finite element model includes or can include analysis of cerebellum and brainstem. Data on brainstem and cerebellum should be supplied as they are significant areas of innervation to and projections from the areas under study. Further, they are intimately involved in both righting reflex and locomotion and may lend to the interpretation of this and future studies.

Figure 3C: Similar to Figure 1, provide outlines of regional anatomy and quantitation of TH vs "other cells". For 3C-F, baseline results should be reported.

Figure 4a: Include latency information.

Figure S1: Provide information and details/controls for Panel F (see reviewer 3 comment).

Methods Revisions:

Details in the methods section need to be better described throughout, examples of which include:

– The *C. elegans* blunt force trauma model is not well described in the methods section. Additionally, since N. Egge et al., 2021 already performed an RNAi screen of dopaminergic neuron GFP retention following injury (but used a different quantification metric), it would be informative to know how strong the effect of cox5b RNAi is in comparison to the top hits in the previous screen.

– In Figure S1C, how were the neurons counted? Is there a co-stain?

– In Table 1, the material properties of rat brain regions are listed, is this a fair comparison to make while modeling the mouse brain?

– Surf KO mice were described to have rotarod defects in C. Dell'agnello et al., 2007, so in figure 3E and F, it would be useful to know the baseline levels of performance prior to injury.

– Unclear how the experiment in Figure S3E was performed, and the conclusions derived from this experiment are also confounding.

– Single nuclei isolation, purification, and RNA sequencing: The reported dissection method will produce single nuclei representing subcortical / diencephalic structures, midbrain, pontine and medullary tissues. Interpretation of snRNAseq results cannot therefore be attributed to represent midbrain or striatal cell populations.

– Statistical Analysis: It is inadequate to state "Post-hoc analysis was determined by Prism recommendation". Please provide details on the types and specific uses of post-hoc analyses.

– Motor Analysis: What make, model and manufacturer is the Rotorod being reported? On the testing day, how many times were the mice tested and does the reported "latency to fall" represent the average? If it does not represent the average, what does it represent?

– Confocal imaging: Was a single image plane acquired or were they z-stack images? How were the images controlled between subjects / slices? If they were z-stacks, what was the step size? How were the images displayed in the manuscript- were they single planes or z-stacks? Were all the image volumes identical?

– Protein extraction and western blotting: Reported methods do not appear to include the use of a phosphatase inhibitor. Therefore western blots showing any changes in phosphorylation may be related more to target protein abundance and not differential phosphorylation. Westerns blots therefore ought to include probes against the non-phosphorylated "total" target protein then in order to help interpret apparent expression changes.

– Media acidification assay: Product numbers and formulation of DMEM is critical for the interpretation of the reported method. At present the method description appears inadequate to provide the type of data desired or reported. Further, cell counts must be performed after the assay is performed since mitochondrial mutations are known to alter population growth kinetics. Therefore acidification results ought to be normalized against population numbers to control for this confound and allow an accurate interpretation of the results.

– Large-particle flow cytometry: How were data calibrated and normalized?

– Closed-head traumatic brain injury: What % isoflurane, what flow rate oxygen, were the mice connected to isoflurane when impacted or not, was the scalp intact (as assumed to be but not explicitly stated)?

– Provide additional experimental details and OCR profiles for Figure 4C OCR measurement.

*Reviewer #1:*

Fonseca et al., cleverly utilize *C. elegans* and mouse models of blunt force trauma to present a elcohesive picture of the signaling pathways underlying neurodegeneration following concussion.

The first major conclusion of the paper is that trauma induced neurodegeneration is neuron subtype specific, with dopaminergic neurons being the most susceptible after injury. This claim is convincingly validated in a *C. elegans* model of blunt force trauma that they previously developed (N. Egge et al., 2021). The authors use a weighted drop model of traumatic brain injury in mice to find a similar hypersensitivity to neurodegeneration in dopaminergic midbrain neurons, but it is unclear if this is specific as other relevant brain areas like the thalamus and cortex have not been assayed.

The authors also identify a key neuroprotective role for ETC complex IV inhibition following injury. Inhibition of ETC Complex IV results in an astrocyte mediated Warburg like shift in metabolism from oxidative phosphorylation to glycolysis, which is sufficient to counteract an increase in toxic ROS production in injured neurons. These observations are reached using an impressive array of experimental techniques, ranging from metabolomic profiling to single nuclei RNA sequencing, with the authors providing multiple lines of evidence in both *C. elegans* and mouse models in support of these novel findings.

Additional experiments:

– Caspase 3 staining following injury must be done for thalamus and cortex of injured mice to show that the neurodegeneration is midbrain dopaminergic neuron specific, rather than just the hippocampus, which is farther than the midbrain from the site of impact.

– Is the Warburg shift in *C. elegans* dependent on non-neuronal cells? If possible, could you perform cell specific RNAi for cox5b and pdp-1 in neurons vs glial sheath cells or the epidermis to test?

Experimental methods:

– The *C. elegans* blunt force trauma model is not described in the methods section. Additionally, since N. Egge et al., 2021 already performed an RNAi screen of dopaminergic neuron GFP retention following injury (but used a different quantification metric), it would be informative to know how strong the effect of cox5b RNAi is in comparison to the top hits in the previous screen.

– Representative images of the GABAergic, cholinergic and serotonergic *C. elegans* neurons following injury would be helpful.

– In Figure S1C, how were the neurons counted? Is there a co-stain?

– Image quality is poor in Figure S1E.

– In Table 1, the material properties of rat brain regions are listed, is this a fair comparison to make while modeling the mouse brain?

– Surf KO mice were described to have rotarod defects in C. Dell'agnello et al., 2007, so in figure 3E and F, it would be useful to know the baseline levels of performance prior to injury.

– Unclear how the experiment in Figure S3E was performed, and the conclusions derived from this experiment are also confounding.

– A major experimental choice that is not explained is the reason why two different components of complex IV are used in *C. elegans* and mouse.

– Unclear about what is being compared in Figure 4F.

– Label uninjured vs injured in Figure 5 E and F.

Writing:

– In the introduction section, a paragraph summarizing your findings would be helpful in orienting the reader to the relevance of the work in relation to previous studies.

– There are a lot of references to Parkinsons disease mediated dopaminergic neurodegeneration, but the link to the current findings is somewhat tenuous, especially as the behavioral deficits observed in the mouse injury model are only a small subset of the motor and cognitive deficits seen in Parkinsons models.

*Reviewer #2:*

This study identified a neuroprotective mechanism using experimental models of blunt force trauma in *C. elegans* and concussion in mice. Interestingly, reducing Mitochondrial CIV function elevates mitochondrial-derived reactive resulted in neuroprotective effect via ROS and HIF1 mediated metabolic transition. Mechanism studies on ROS and metabolic regulation are interesting. The worm and mouse models are well established. The mouse model to reduce rather than ablate cytochrome C oxidase activity is a good choice.

1. Line 124, the authors may add a few more words to describe how cox-5b is identified, as a small-scale screening has been performed.

2. Figure 3C-F, baseline results should be provided.

3. The authors may need to determine whether nuo-2, sdhd-1, cyc-1, atp-3 RNAi treatments reduce ROS. Not sure if ROS suppression is cox-5b RNAi specific.

4. If there is a selective vulnerability of different neural subtypes, results using MEFs in figure 4 may not explain the "selective vulnerability" phenotype. Are dopaminergic neurons hypersensitive to ROS after brain injury? The mechanisms need to be better explained.

5. In figure 5, if HIF1 is the "key" regulator, does HIF1 RNAi prevent glycolytic preconditioning and neuroprotection? Similarly, does hypoxia exhibit neuroprotective effect?

*Reviewer #3:*

Traumatic brain injury (TBI) is a significant source of global behavioral, neurological and neurodegenerative morbidity. Though risk factors for poor outcomes following TBI, such as those thought to precipitate Alzheimer's disease are intensely studies, conserved mechanisms of injury and resiliency factors with the potential to prevent poor outcomes remain largely unknown. In this study by Fonseca and colleagues, the study team leverages a combination of model systems (e.g. *C. elegans*, mice, cell systems) to identify the selective vulnerability of dopaminergic neurons following physical trauma. Through a series of well-reasoned and articulate experiments, the study team presents evidence that suppression of function at the mitochondrial electron transport chain complex IV may reduce the loss of dopaminergic neurons following TBI, in part by driving astrocyte-mediated lactate shuttling to reduce oxidative phosphorylation in vulnerable neurons. This is potentially a very significant finding with wide ranging implications not only for acute injury care but potentially chronic neurodegenerative disease as well. A clear strength of the study is its approach to identifying evolutionarily-conserved mechanisms of selective dopaminergic vulnerability to TBI, which will likely be critical to identifying actionable targets in human TBI. The study is well written with an interesting and thoughtful narrative. The weakness of the paper is primarily the under-reporting of methodological details essential to reproducing the work, inadequate controls to verify results, and a paucity of data supporting major claims and conclusions.

Recommendations for the authors:

Line 46: Unclear statement. Clarify.

Line 82: Change "degeneration" to "injury".

Line 97 to 100: TBI-induced deficits in voluntary movement are well established and may have multiple causes unrelated to SN / DA injury. Interestingly, changes in "disinhibition and risk seeking", another classical Parkinsonian trait and more cleanly associated with dopaminergic dysfunction, has been reported in TBI in human subjects (self-report) and mouse TBI models (using novel object recognition tests). Provide results of novel object recognition tests to demonstrate unencumbered behavioral / functional impairment related to dopaminergic injury / degeneration in this mouse model. Evaluate the correlation between behavioral results with the degree of dopaminergic loss across SNr, SNc, VTA, etc.

Line 113-115: Caution and conservativeness is warranted in this closing statement. This study has not adequately compared and contrasted the profiles of neuronal injury across the SNc/Midbrain with those of the thalamus and therefore its inappropriate to suggest that the inherent physiological properties of dopaminergic neurons sensitizes them to biomechanical insults [as compared with the thalamus]. Indeed several studies have noted substantial thalamic alterations associated with repeated TBI, whereas few have identified dopaminergic pathology.

Line 139: As written, your statement makes it sound as though though evolution has selected for the protection of DA neurons against TBI by reduction of cytochrome c oxidase activity, which I do not think you intend to convey. Perhaps, "suppressing trauma-induced degeneration of dopaminergic neurons through reduction in cytochrome C oxidase activity is sufficient to conserve dopaminergic function in both *C. elegans* to mice."?

Major concern Line 150 to 151: If I am understanding correctly… As written, worms are injured and then placed into RNAi lawns to suppress de novo translation of cox5b to the effect of mitigating ROS production. However, this does not make sense, since ROS production should be able to be generated with existing mitochondrial protein complexes and not necessarily require new protein. This would suggest the vast literature of "tired old mitochondrial proteins are the cause of ROS" is collectively wrong. Resolve this apparent conflict. Possible methods may use cycloheximide to halt general translation or use of tet on/off systems controlling target protein expression.

Line 159 to 161: Unclear / confusing. For clarity, please reword or break into separate sentences.

Line 175: Was a correlation statistically evaluated? If so, please report results. If not, change wording. Perhaps use "corresponded".

Line 190: "correlated". Same comment as Line 175.

Line 203: typo "Pdk2". Change to "pdk2" (standard gene symbols are all lower case by convention). Replace "activation" with something more precise such as "upregulation" or "increased transcription". The word activation implies signaling or physiological response.

Line 204: typo "Pdp1". Change to "pdp1" (standard gene symbols are all lower case by convention).

Line 212: Change "spatial" to "cellular". snRNAseq provides cellular data and though cellular nuclei are physically separate from one another, snRNAseq does not preserve accepted information on spatial relationships such as that provided by microscopy.

Line 219 to 221: "Through spatial resolution…" Identifying a pro-Warburg shift in astrocytes driven by Surf1 KO, does not demonstrate that this is a pathway for naturally occuring Warburg phenomenon. Suggest, softening by changing text to read "…may be initiated by electron…". Further, though the presented data indicates the Warburg shift is princpally manifest in astrocytes, it does not rule out the initiation of this process by extrinsic factors such as exosomes.

Line 229: Hif1alpha… gene or protein?

Line 236: Hif1alpha… gene or protein?

Line 239: Hif1alpha… gene or protein?

Line 245: change "pdp-1" to "pdp1".

Line 249: change to "… trauma-induced neurodegeneration of dopaminergic neurons…"

Table 1.

Are these parameters for mouse brain or rat brain? If they are for rat, then provide the correct parameters for mouse brain. If the title is in error, then correct the title.

Provide a descriptor as to what each parameter is and what higher or lower numbers indicate for the lay reader.

Figure 1.

Panel C: Requires a reference map or annotations directly on the image to determine the precise anatomical location(s) pictured in the micrographs in order to inform the reader as to the principle cell populations involved. If this is the midbrain Substantia nigra, please label SNp, SNr, VTA, etc. If this is not precisely in that area, additional tissue staining may be required to identify cell types. TH (tyrosine hydroxylase) is a marker of catecholaminergic cells and not specifically a marker of dopaminergic cells. Staining with TH will therefore include dopaminergic and noradrenergic cells which vary in their distribution throughout the midbrain to pons. If needed use dopamine β-hydroxylase (DBH) in addition to TH to differentiate dopaminergic cells from noradrenergic cells.

Use of cleaved caspase-3 staining is at insufficient resolution to determine whether TH cells (or an unrelated phenotype) are positive for cleaved caspase-3. Quantitation of colocalized cleaved caspase-3 across TH^+^, astrocytes, vs other cell types should be supplied.

Use of additional markers are required to verify the apoptotic cells are dopaminergic neurons and not simply other cell types such as infiltrated peripheral immune cell subsets (which are known to express tyrosine hydroxylase and apoptosize during the early phases of brain injury resolution).

Panel C: At what time point?

Panel D: At what time point? How many slices per animal were examined?

Panel E: It is unusual for uninjured mice to fall off the rotorod so quickly, especially when they were previously trained on the task two days prior. Were the animals placed on the rod "backwards"? Provide data for all tested speeds if available.

Figure 2.

Between A and B: There is an illustrated "block". I am assuming this is either an illustrated "typo" or the impactor head. Please define or remove this "block".

C. Both Table 1 and Figure S2 indicate the finite element model includes or can include analysis of cerebellum and brainstem. Data on brainstem and cerebellum should be supplied as they are significant areas of innervation to and projections from the areas under study. Further, they are intimately involved in both righting reflex and locomotion and may lend to the interpretation of this and future studies.

Figure 3.

Panel A and B: At what time point? Also, provide either quantitative real-time PCR or Western blot quanitation of relative RNAi efficiency.

Panel C: Similar to Figure 1, provide outlines of regional anatomy and quantitation of TH vs "other cells" costaining along with "zoomed in" insets to verify colocalization of cleaved caspase-3 with dopaminergic neurons.

Panel C: At what time point are the brain slices representing?

Panel D: Are these TBI or control mice?

Panel F: Are these TBI or Control mice? If TBI, how long after TBI are the mice given this test?

Figure 4.

Title: Change "Reducing cytochrome C…" to "Decreasing cytochrome C oxidase expression…". This helps avoid confusion between electrochemical "reductions" involving cytochrome C oxidase and changes in cytochrome C protein expression levels, such as the use of RNAi intends to cause.

Panel A: Was there no latency between when the trauma occured and when they began peroxide measurements? If so, include that in the timescale.

Panel C: Data lines are unlabeled as to the reagent used to test OCR. Graph should consist of control, actinomycin, FCCP and Oligomycin treatment conditions for each genotype. It is impossible to conclude that this even has living cells without providing more data.

Recommend using a dynamic Seahorse ECAR / OCR assay. Please provide the full Seahorse Glycolytic ECAR and Mitochondrial respiratory OCR profiles.

Panel E: n=3. Is this technical or biological replicates? How was this measured (mention in legend)?

Figure 5.

Panel B: Please run all samples on a single western blot and show results as a single boxed area. When western blots are presented such as this it is difficult to determine an effect of "injury". As it is, the reader cannot conclude that there's any difference between conditions other than a main effect of genotype.

Figure Legend Text for Panel C: remove the word "neuronal" or replace with "neural".

Panel C (figure): Gene symbols are not capitalized by convention.

Panel F: Requires single cell quantitation across DA neurons, astrocytes, others to determine treatment effects differentially affect apoptosis in a partcular target cell type.

Panel G and H: Change pdp-1 to pdp1.

Figure S1.

Panel C: How many slices per mouse were examined?

Panel D and E: What study timepoint do these micrographs represent? How many slices per mouse were examined?

Panel F: It is unusual to detect no change across a number of microglia enriched genes in areas that are experiencing neuronal apoptosis, especially in CD68 which would be under greater use in the elimination of cellular debris. How was this data normalized? Was it library normalized per cell? If not, consider that analytical approach. What type of RNAseq and where was the material collected from? (I am assuming snRNAseq from the described subdissection) Provide positive and negative controls to prove the assay worked. Ensure the microglia being reported on are from the area under study and not from a broadly mixed tissue source, which may have originally come from "uninjured" locations by sheer random chance and may not represent the SN. Microscopy or morphometric analyses may be required to ensure proximity.

Figure S2.

No comments.

Figure S3.

Panel A: Are these Injury or control mice?

Panel B: Mortality should be represented as a curve across increasing impact force. For a first demonstration of mortality curves, ideally data from at least three different cohorts, with each cohort injured separately from one another on different days, should be supplied to allow the reader to determine the degree of reproducibility for both the execution of the method and the experimental measure being made.

Panels D and E: Panel D shows an apparent loss of DA neurons in injured mice with a compensatory upregulation of TH by remaining neurons 28 days following injury. Panel E quantifies the DA neuron loss for both Surf wt and KO mouse strains. What is the average immunofluorescent expression of TH by remaining DA neurons after injury and does it differ by strain?

Figure S4.

No comments

Figure S5.

Panel A: Are these control mice?

Panel B to E: At what timepoint after injury does this data represent?

Panel F and G: these panels appear to be misidentified in the figure legend. Please check.

Panel M: It does not seem reasonable to acheive a snRNAseq of midbrain and striatal brain tissue without obtaining microglia and endothelial cells. Please reconcile this and report these cell types.

Methods:

Single nuclei isolation, purification, and RNA sequencing: The reported dissection method will produce single nuclei representing subcortical / diencephalic structures, midbrain, pontine and medullary tissues. Interpretation of snRNAseq results cannot therefore be attributed to represent midbrain or striatal cell populations.

Closed-head traumatic brain injury: What % isoflurane, what flow rate oxygen, were the mice connected to isoflurane when impacted or not, was the scalp intact (as assumed to be but not explicitly stated)?

Media acidification assay: Product numbers and formulation of DMEM is critical for the interpretation of the reported method. At present the method description appears inadequate to provide the type of data desired or reported. Further, cell counts must be performed after the assay is performed since mitochondrial mutations are known to alter population growth kinetics. Therefore acidification results ought to be normalized against population numbers to control for this confound and allow an accurate interpretation of the results. Confirmation of acidification assay should also be performed using a secondary method such as pH meter.

Statistical Analysis: It is inadequate to state "Post-hoc analysis was determined by Prism recommendation". Please provide details on the types and specific uses of post-hoc analyses.

RNAi administration: RNAi sequences and suppliers and concentrations must be provided.

Large-particle flow cytometry: How were data calibrated and normalized?

Motor Analysis: What make, model and manufacturer is the Rotorod being reported? On the testing day, how many times were the mice tested and does the reported "latency to fall" represent the average? If it does not represent the average, what does it represent?

Confocal imaging: Was a single image plane acquired or were they z-stack images? How were the images controled between subjects / slices? If they were z-stacks, what was the step size? How were the images displayed in the manuscript- were they single planes or z-stacks? Were all the image volumes identical?

Protein extraction and western blotting: Reported methods do not appear to include the use of a phosphatase inhibitor. Therefore western blots showing any changes in phosphorylation may be related more to target protein abundance and not differential phosphorylation. Westerns blots therefore ought to inculde probes against the non-phosphorylated "total" target protein then in order to help interpret apparent expression changes.

General comment: The study generally uses n=3 mice per group for many experiments. n=3 subjects per group are typically only acceptable with rare materials / subjects. I would argue that n=3 is inadequate for any mouse study.

---

## [Author Response]

Additional experiments:1. Caspase 3 staining following injury should be done for thalamus and cortex of injured mice to show that the neurodegeneration is midbrain dopaminergic neuron specific, rather than just the hippocampus, which is farther than the midbrain from the site of impact.

We have included new indirect immunofluorescence micrographs and quantifications for the respective brain regions (thalamus and cortex). This additional data complements our existing analysis of the midbrain and hippocampus. In brief, there was no significant change in immunostained cleaved caspase 3 signal intensity seven days post-concussion in both the thalamus or the cortex. These new data are included in the revised manuscript in Figure 1—figure supplement 1.

2. Clarification or modification/retesting of the worm cox5b RNAi protocol: As written, worms are injured and then placed into RNAi lawns to suppress de novo translation of cox5b to the effect of mitigating ROS production. However, this does not make sense, since ROS production should be able to be generated with existing mitochondrial protein complexes and not necessarily require new protein. This would suggest the vast literature of "tired old mitochondrial proteins are the cause of ROS" is collectively wrong. Resolve this apparent conflict. Possible methods may use cycloheximide to halt general translation or use of tet on/off systems controlling target protein expression.

To resolve the confusion regarding the timeline of RNAi administration and injury, we have included additional text in the revised methods section on worm trauma. All RNAi treatments are applied to the animal throughout their life. Thus, animals have been preconditioned prior to the injury. For *cox5b* RNAi, animals have reduced translation of the COX5b protein leading up to the injury. In this case, we proposed that reduced reliance on mitochondria for energetics mitigates ROS production resulting from an overwhelmed ETC.

3. Restaining and/or reimaging figure 1C (see below).

We appreciate the recommendation. We have taken the advice of the reviewer and labeled the different regions of the midbrain (1, 2 and 3), which are now described in more depth in the figure legend of the revised manuscript.

4. Panel 5B: Please run all samples on a single western blot and show results as a single boxed area. When western blots are presented such as this it is difficult to determine an effect of "injury".

All protein extracts were resolved on a single western blot and included in Figure 5—figure supplement 2 of the revised manuscript. Whether prior to (indicating a preconditioning) or at the onset of trauma-induced dopaminergic apoptosis (day 7 post injury), SURF1 mutant animals consistently show elevated PDHE1 phosphorylation, suggesting that they are favoring cytosolic glycolysis prior to the injury or coming out of the injury. This timing is consistent with previous reports, which examined the effects of open-head, controlled cortical impact (CCI)^1^.

Textual revisions: I strongly recommend reading reviewer 3's thorough review and verifying/modifying all of the typographical / nomenclature/ clarity changes requested.

We appreciate the thorough review and have carefully gone through all the recommended corrections as stated below.

1. Line 46: Unclear statement. Clarify.

We have adjusted the text in the revised manuscript to simplify the message.

2. Line 113-115: please temper statement.

We have reworded the respective sentence to temper the statement as a hypothesis (now line 122-125).

3. Line 139: modify for clarity.

After trying a few different versions of this concluding sentence (now line 155-157), we decided to keep the original text but are happy to entertain alternate text suggestions for this process being conserved.

4. Line 159 to 161: modify for clarity.

In the revised text, we reworded the respective sentence to clarify the epistasis analysis (now line 177-179).

5. Line 219 to 221: temper statement.

This text was adjusted to state “we provide evidence” rather than a definitive statement (now line 244-246).

6. Line 82: Change "degeneration" to "injury".

We changed the respective “neurodegeneration” to “neural damage”.

7. Line 175/190: Was a correlation statistically evaluated? If so, please report results. If not, change wording. Perhaps use "corresponded".

We have adjusted the text to say “corresponded”.

8. Line 203: Replace "activation" with something more precise such as "upregulation" or "increased transcription". The word activation implies signaling or physiological response.

We have adjusted the text to say “upregulation” versus “activation”.

9. Line 212: Change "spatial" to "cellular".

We have made the respective change in the revised manuscript.

10. Line 219 to 221: soften claim.

We have included the respective text “our data suggests that” to soften the claim and state that this is not a definitive statement but rather a suggestion.

Figure/Table Revisions:Please provide timepoints for all relevant figures where not listed. Also please carefully label control and TBI mice in each panel where relevant, and provide n where not clear.Table 1. Better describe and clarify if parameters are for mouse brain or rat brain. If they are for rat, then provide the correct parameters for mouse brain. If the title is in error, then correct the title.

In the original manuscript, parameters were defined for the rat brain. Unlike the rat brain, viscoelastic properties for some mouse brain regions (the hippocampus in particular) have not been defined or reported in the literature. To address this, we performed the respective analysis with updated mouse brain parameters with the exception of the hippocampus. In this case, we superimposed the viscoelastic parameters of the rat hippocampus into the mouse model. These new properties for the revised model have been updated in Table 1 and additional text included in the revised methods section. Regardless of the rat brain injury model or the mouse/rat hybrid model, the conclusion remains the same. TBI promotes a wave of mechanical strain, which propagates through the whole brain.

Figure 1. Panel C: Requires a reference map or annotations directly on the image to determine the precise anatomical location(s) pictured in the micrographs. If this is the midbrain Substantia nigra, please label SNp, SNr, VTA, etc. If this is not precisely in that area, additional tissue staining may be required to identify cell types.

In the revised manuscript, we have included labels (1, 2 and 3) in the Figure to highlight the respective regions of the midbrain.

TH (tyrosine hydroxylase) is a marker of catecholaminergic cells and not specifically a marker of dopaminergic cells. Staining with TH will therefore include dopaminergic and noradrenergic cells which vary in their distribution throughout the midbrain to pons. If needed use dopamine β-hydroxylase (DBH) in addition to TH to differentiate dopaminergic cells from noradrenergic cells.

It is worth noting that any cell with TH will synthesize dopamine. However, noradrenergic cells convert the dopamine to norepinephrine and this occurs in the locus coeruleus, which is not in the brain section that we examined. Although noradrenergic cells invert part of the midbrain, no NA cell bodies have been reported in this brain region^2^.

Panel E: It is unusual for uninjured mice to fall off the rotorod so quickly, especially when they were previously trained on the task two days prior. Provide data for all tested speeds if available.

We did not use a rotarod training paradigm. Rather we sought to perform a more naïve motor coordination analysis. In brief, mice were placed in the chamber 2 days prior to injury without turning on the machine. This time was used to familiarize the mice with their surroundings and enabled us to get more consistent readings after trauma. We have included additional text in the revised methods section to better clarify the assay and analysis.

Both Table 1 and Figure S2 indicate the finite element model includes or can include analysis of cerebellum and brainstem. Data on brainstem and cerebellum should be supplied as they are significant areas of innervation to and projections from the areas under study. Further, they are intimately involved in both righting reflex and locomotion and may lend to the interpretation of this and future studies.

This is a good point and we have now included analysis on the additional brain regions (cerebellum and brainstem) in the revised text.

Figure 3C: Similar to Figure 1, provide outlines of regional anatomy and quantitation of TH vs "other cells".

In the revised manuscript, we have included labels in the figure to mark all the respective regions of the midbrain (1, 2 and 3).

For 3C-F, baseline results should be reported.

We have included baseline readings for the respective non-injured animals in the corresponding supplemental figures.

Figure 4a: Include latency information.

We are uncertain what latency information is being requested for H2O2 production. Both control and *cox5b* RNAi treated animals are provided with and without injury. All animals are age matched.

Figure S1: Provide information and details/controls for Panel F (see reviewer 3 comment).

The respective transcriptomic data was from RNAseq analysis of the whole brain. In this case, abundance for each inflammatory-linked transcript is normalized to reads per kilobase million (RPKM) as we have done in previous publications^3,4^. For ease of viewing, we have made age-matched control transcript levels relative to 1 and adjusted injured timepoints (2 hours and 7 days) accordingly. We have included additional text in the methods section to explain this normalization by RPKMs. Since this is a new method of closed-head injury, which does not immediately compromise the blood-brain barrier, many of the transcriptional targets that might serve as positive controls are not applicable since we did not perform the craniotomy surgery. Moreover, age-matched control animals which did not receive the injury were still anesthetized with isoflurane and allowed to recover. All these details are described in the methods section of the revised manuscript.

Methods Revisions:Details in the methods section need to be better described throughout, examples of which include:– The *C. elegans* blunt force trauma model is not well described in the methods section. Additionally, since N. Egge et al., 2021 already performed an RNAi screen of dopaminergic neuron GFP retention following injury (but used a different quantification metric), it would be informative to know how strong the effect of cox5b RNAi is in comparison to the top hits in the previous screen.

In the Egge et al., 2021 *Nature Communication* manuscript, we utilized both GFP retention within the dopaminergic neurons (the quantification metric utilized in the present manuscript under review) as well as the trauma index. In this previous publication^3^, the trauma index was formulated for ease of comparisons across the various RNAi conditions (over 50 conditions). With regards to *cox-5b* RNAi, we observe a trauma index of 0.70 compared to what we previously reported for *vhp-1* RNAi with an index of 0.75. Thus, *cox-5b* RNAi is comparable to *vhp-1* RNAi with regards to providing robust neuroprotection for dopaminergic neurons in *C. elegans* after blunt force injury.

– In Figure S1C, how were the neurons counted? Is there a co-stain?

The revised method section provides more in-depth detail for how we collected and quantified confocal micrographs. In brief, TH positive cells were co-stained with DAPI and Nissl prior to counting and comparisons across groups.

– In Table 1, the material properties of rat brain regions are listed, is this a fair comparison to make while modeling the mouse brain?

As describe above, the viscoelastic properties for the mouse brain regions (aside from the hippocampus) were used in the revised brain injury model. Overall, these minor changes in viscoelasticity between rat and mouse provide similar results in the TBI model.

– Surf KO mice were described to have rotarod defects in C. Dell'agnello et al., 2007, so in figure 3E and F, it would be useful to know the baseline levels of performance prior to injury.

In the revised manuscript, we include baseline readings for naïve mice both wild type and *Surf1* mutants. Although C. Dell’agnello et al., 2007 report rotarod defects in *Surf1* mutants, we do not observe significant differences in latency between wild type animals and the mutants. It is worth noting that our laboratory utilizes a different rotarod paradigm, making it difficult to draw comparisons between the different experimental setups. These baseline readings are included in the revised manuscript in Figure 3—figure supplement 1.

– Unclear how the experiment in Figure S3E was performed, and the conclusions derived from this experiment are also confounding.

Confocal images were collected and quantified as performed in Figure 1—figure supplement 1C. Neuronal counts in the substantia nigra were compared across groups and statistically analyzed following the details in the methods section of the manuscript. The conclusions from this experiment show enhanced survival of dopaminergic neurons in the *Surf1* mutants 28 days after closed head injury. This data corresponds with *Surf1* mutant animals showing significantly less cleaved caspase 3 staining in the same brain region 7 days post-concussion.

– Single nuclei isolation, purification, and RNA sequencing: The reported dissection method will produce single nuclei representing subcortical / diencephalic structures, midbrain, pontine and medullary tissues. Interpretation of snRNAseq results cannot therefore be attributed to represent midbrain or striatal cell populations.

This is a good point and exactly why we leveraged single nuclei sequencing techniques. In collaboration with established experts on single cell analysis (laboratory of Dr. Konopka), we were able to distinguish dopamine producing cells from other cell types and analyze transcriptional differences within the defining sub-groups or “clusters” via genetic profiling (Figure 5—figure supplement 1N).

– Statistical Analysis: It is inadequate to state "Post-hoc analysis was determined by Prism recommendation". Please provide details on the types and specific uses of post-hoc analyses.

In the methods of the revised text, we have included more extensive details regarding our Post-hoc analysis. In brief, student’s t-test was used to compare means between two normal populations. Mann-Whitney U test was used to compare differences in the dependent variable between two groups. Post-hoc analysis performed after ANOVA included Dunnett’s multiple comparison (to compare means from several experimental groups against a control group mean) and Tukey’s multiple comparisons test (to compare all possible pairs of means). These details are provided in the revised figure legends.

– Motor Analysis: What make, model and manufacturer is the Rotorod being reported? On the testing day, how many times were the mice tested and does the reported "latency to fall" represent the average? If it does not represent the average, what does it represent?

In the methods section of the revised manuscript, we have included the make, model, and manufacturer of our laboratory’s rotarod. Moreover, we provide a detailed description of the procedure. In brief, motor deficits in mice were measured by Rotarod (Cat. No. 76-0770; Harvard Apparatus) with a rod diameter of 3 cm and a rod height of 20 cm. Animals were not trained on a rotating rod prior to testing, but rather exposed to the Rotarod machine chamber for 2 minutes (3 times daily). For testing, animals were placed on the rod with accelerating speed (0-24 rpm in 120 s) for all experiments. Latency to fall was recorded electronically in seconds by the apparatus and values were averaged per group for reporting. The clock was stopped if an animal held to the rod on two consecutive rotations and if the animal failed to fall after 120 s. Animals were returned to their cage after each trial.

– Confocal imaging: Was a single image plane acquired or were they z-stack images? How were the images controlled between subjects / slices? If they were z-stacks, what was the step size? How were the images displayed in the manuscript- were they single planes or z-stacks? Were all the image volumes identical?

In the methods section of the revised manuscript, we have included additional text to provide more details regarding the confocal microscopy. Briefly, brain parenchyma images were collected as Z-stacks with 1 µm steps as follows: midbrain micrographs 20 µm/10x objective/5 tile Z-stacks, cortex micrographs 10 µm/40x objective/5 tile Z-stacks, hippocampus micrographs 20 µm/10x objective/8 tile Z-stacks, thalamus micrographs 5 µm/40x objective/2 tile Z-stacks. Number of tiles to be collected was determined by completely covering the brain structure of interest (i.e. midbrain 1x5 tiles, hippocampus 2x4 tiles). Tiled images were stitched with the Leica LAS X software. Cell culture micrographs were collected from cells grown on glass coverslips with a total of 10 fields collected with a 40x objective for image processing and analysis. Brain parenchyma images are displayed as maximal intensity projections of Z-stacks. Cell culture images are displayed as single plane micrographs.

– Protein extraction and western blotting: Reported methods do not appear to include the use of a phosphatase inhibitor. Therefore western blots showing any changes in phosphorylation may be related more to target protein abundance and not differential phosphorylation. Westerns blots therefore ought to include probes against the non-phosphorylated "total" target protein then in order to help interpret apparent expression changes.

All cell extracts were created in the presence of the SDS detergent, which will linearize folded proteins and inactivate enzymes including phosphatases and kinases. Even alkaline phosphatases, which are known to be resistant to lower concentrations of SDS, can be inactivated at 1% SDS^5^, which is the concentration used in our lysis buffer. This point has been added to the revised methods section. Regardless, we have now included additional western blots of total PDH and confirm that its steady-state levels are not significantly altered in the different conditions.

– Media acidification assay: Product numbers and formulation of DMEM is critical for the interpretation of the reported method. At present the method description appears inadequate to provide the type of data desired or reported. Further, cell counts must be performed after the assay is performed since mitochondrial mutations are known to alter population growth kinetics. Therefore acidification results ought to be normalized against population numbers to control for this confound and allow an accurate interpretation of the results.

In the revised methods section of the manuscript, we provide additional details regarding this particular assay. We specify the use of DMEM/F-12 (Cat. No. 11320033; Gibco), which contains phenol red with a pH indicator ranging of 6.8 to 8.2 according to the Merck Index, 13^th^ edition, 7329. Methods used in the paper were adapted from a 2018 Resource-Application note from BioTek titled, “Using phenol red to assess pH in Tissue culture media”. The link is provided below. Moreover, data was normalized to 1x10^5^ cells for analysis.

(https://www.biotek.com/resources/application-notes/using-phenol-red-to-assess-ph-in-tissue-culture-media/).

– Large-particle flow cytometry: How were data calibrated and normalized?

Prior to every run, laser power and flow rates are calibrated by use of GP control particles (COPAS Biosorter, 310-5071-001) as recommended by the company. From experiment to experiment, we maintain a consistent PTM laser power for each respective fluorophore. Moreover, we run an empty vector (EV) and control (both injured and non-injured) which we use to normalize all our experimental values.

– Closed-head traumatic brain injury: What % isoflurane, what flow rate oxygen, were the mice connected to isoflurane when impacted or not, was the scalp intact (as assumed to be but not explicitly stated)?

In the methods section of the revised manuscript, we provide all these specific details. In brief, we use 3% USP grade isoflurane and 2.5 lpm USP grade oxygen. The procedure takes under 10 seconds to set up the mice and administer the injury, so it is not necessary for the mice to remain connected to isoflurane. The scalp remained intact on these mice.

– Provide additional experimental details and OCR profiles for Figure 4C OCR measurement.

Since we did not use a Seahorse efflux analyzer to determine OCR, no OCR profiles (which are typical outputs for the Seahorse) were obtained. Rather, we used a commercially available kit (Cayman Chemicals) to determine OCR measurement. This and catalog numbers for the reagents are now clearly stated in the revised methods section.

References

1. Watson, W. D. *et al.,* Impaired cortical mitochondrial function following TBI precedes behavioral changes. *Front Neuroenergetics* 5, 12, doi:10.3389/fnene.2013.00012 (2013).

2. Jenkins, P. O., Mehta, M. A. and Sharp, D. J. Catecholamines and cognition after traumatic brain injury. *Brain* 139, 2345-2371, doi:10.1093/brain/aww128 (2016).

3. Egge, N. *et al.,* Trauma-induced regulation of VHP-1 modulates the cellular response to mechanical stress. *Nat Commun* 12, 1484, doi:10.1038/s41467-021-21611-8 (2021).

4. Egge, N. *et al.,* Age-Onset Phosphorylation of a Minor Actin Variant Promotes Intestinal Barrier Dysfunction. *Dev Cell* 51, 587-601 e587, doi:10.1016/j.devcel.2019.11.001 (2019).

5. Stinson, R. A. Size and stability to sodium dodecyl sulfate of alkaline phosphatases from their three established human genes. *Biochim Biophys Acta* 790, 268-274, doi:10.1016/0167-4838(84)90031-1 (1984).

6. Lee, S. J., Hwang, A. B. and Kenyon, C. Inhibition of respiration extends *C. elegans* life span via reactive oxygen species that increase HIF-1 activity. *Curr Biol* 20, 2131-2136, doi:10.1016/j.cub.2010.10.057 (2010).

7. Hekimi, S., Wang, Y. and Noe, A. Mitochondrial ROS and the Effectors of the Intrinsic Apoptotic Pathway in Aging Cells: The Discerning Killers! *Front Genet* 7, 161, doi:10.3389/fgene.2016.00161 (2016).

8. Crane, P. K. *et al.,* Association of Traumatic Brain Injury With Late-Life Neurodegenerative Conditions and Neuropathologic Findings. *JAMA Neurol* 73, 1062-1069, doi:10.1001/jamaneurol.2016.1948 (2016).

Reviewer #1:Fonseca et al., cleverly utilize *C. elegans* and mouse models of blunt force trauma to present a elcohesive picture of the signaling pathways underlying neurodegeneration following concussion.The first major conclusion of the paper is that trauma induced neurodegeneration is neuron subtype specific, with dopaminergic neurons being the most susceptible after injury. This claim is convincingly validated in a *C. elegans* model of blunt force trauma that they previously developed (N. Egge et al., 2021). The authors use a weighted drop model of traumatic brain injury in mice to find a similar hypersensitivity to neurodegeneration in dopaminergic midbrain neurons, but it is unclear if this is specific as other relevant brain areas like the thalamus and cortex have not been assayed.The authors also identify a key neuroprotective role for ETC complex IV inhibition following injury. Inhibition of ETC Complex IV results in an astrocyte mediated Warburg like shift in metabolism from oxidative phosphorylation to glycolysis, which is sufficient to counteract an increase in toxic ROS production in injured neurons. These observations are reached using an impressive array of experimental techniques, ranging from metabolomic profiling to single nuclei RNA sequencing, with the authors providing multiple lines of evidence in both *C. elegans* and mouse models in support of these novel findings.Additional experiments:– Caspase 3 staining following injury must be done for thalamus and cortex of injured mice to show that the neurodegeneration is midbrain dopaminergic neuron specific, rather than just the hippocampus, which is farther than the midbrain from the site of impact.

We have included new indirect immunofluorescence micrographs and quantifications for the respective brain regions (thalamus and cortex). This additional data complements our existing analysis of the midbrain and hippocampus. In brief, there was no significant change in immunostained cleaved caspase 3 signal intensity seven days post-concussion in both the thalamus or the cortex. These new data are included in the revised manuscript in Figure 1—figure supplement 1.

– Is the Warburg shift in *C. elegans* dependent on non-neuronal cells? If possible, could you perform cell specific RNAi for cox5b and pdp-1 in neurons vs glial sheath cells or the epidermis to test?

This is a good point. Transgenic *C. elegans* strains, which enabled tissue specific RNAi exclusively in the nervous system, were used to show that *cox-5b* and *pdp-1* RNAi are both able to suppress trauma-induced loss of dopaminergic GFP. This new data is now included in the Figure 3—figure supplement 1 and Figure 5—figure supplement 2 of the revised manuscript. Attempts to construct a sheath glia-specific RNAi worm strain were not successful. This remains an important question for further studies.

Experimental methods:– The *C. elegans* blunt force trauma model is not described in the methods section. Additionally, since N. Egge et al., 2021 already performed an RNAi screen of dopaminergic neuron GFP retention following injury (but used a different quantification metric), it would be informative to know how strong the effect of cox5b RNAi is in comparison to the top hits in the previous screen.

In the Egge et al., 2021 *Nature Communication* manuscript, we utilized both GFP retention within the dopaminergic neurons (the quantification metric utilized in the present manuscript under review) as well as the trauma index. In this previous publication^3^, the trauma index was formulated for ease of comparisons across the various RNAi conditions (over 50 conditions). With regards to *cox-5b* RNAi, we observe a trauma index of 0.70 compared to what we previously reported for *vhp-1* RNAi with an index of 0.75. Thus *cox-5b* RNAi is comparable to *vhp-1* RNAi with regards to providing robust neuroprotection for dopaminergic neurons in *C. elegans* after blunt force injury.

– Representative images of the GABAergic, cholinergic and serotonergic *C. elegans* neurons following injury would be helpful.

GABAergic images with and without injury were previously reported and analyzed in Egge et al., *Nature Comm*, 2021. Due to no observable changes in fluorescence in cholinergic and serotonergic neurons, we did not include the respective images. It is worth noting that serotonergic neurons are the focus of another paper under review elsewhere.

– In Figure S1C, how were the neurons counted? Is there a co-stain?

The revised methods section provides more in-depth detail for how we collected and quantified confocal micrographs. In brief, TH positive cells were co-stained with DAPI and Nissl prior to counting by eye and comparisons across groups.

– Image quality is poor in Figure S1E.

In the revised manuscript, we have completely reworked the figure and now include high magnification images of the micrograph (now Figure 1—figure supplement 1J).

– In Table 1, the material properties of rat brain regions are listed, is this a fair comparison to make while modeling the mouse brain?

In the original manuscript, parameters were defined for the rat brain. Unlike the rat brain, viscoelastic properties for some mouse brain regions (the hippocampus in particular) have not been defined or reported in the literature. To address this, we performed the respective analysis with updated mouse brain parameters with the exception of the hippocampus. In this case, we superimposed the viscoelastic parameters of the rat hippocampus into the mouse model. These new properties for the revised model have been updated in Table 1 and additional text included in the revised methods section. Regardless of the rat brain injury model or the mouse/rat hybrid model, the conclusion remains the same. TBI promotes a wave of mechanical strain, which propagates through the whole brain.

– Surf KO mice were described to have rotarod defects in C. Dell'agnello et al., 2007, so in figure 3E and F, it would be useful to know the baseline levels of performance prior to injury.

In the revised manuscript, we include baseline readings for naïve mice both wild type and *Surf1* mutants. Although C. Dell’agnello et al., 2007 report rotarod defects in *Surf1* mutants, we do not observe significant differences in latency between wild type animals and the mutants. It is worth noting that our laboratory utilizes a different rotarod paradigm, making it difficult to draw comparisons between the different experimental setups. These baseline readings are now included in the revised manuscript in Figure 3—figure supplement 1.

– Unclear how the experiment in Figure S3E was performed, and the conclusions derived from this experiment are also confounding.

Confocal images were collected and quantified as performed in Figure 1—figure supplement 1C. Neuronal counts in the substantia nigra were compared across groups and statistically analyzed following the details in the methods section of the manuscript. The conclusions from this experiment show enhanced survival of dopaminergic neurons in the *Surf1* mutants 28 days after closed head injury. This data corresponds with *Surf1* mutant animals showing significantly less cleaved caspase 3 staining in the same brain region 7 days post concussion.

– A major experimental choice that is not explained is the reason why two different components of complex IV are used in *C. elegans* and mouse.

This decision was based largely on the available of validated reagents as well as phenotypic consistencies between both worms and mice. Both *cox-5b* RNAi in worms and *Surf1* mutant mice were shown to be beneficial and long-lived. Both are assembly factors for the same ETC complex IV and impairment of either protein, reduces complex IV activity without its full ablation.

– Unclear about what is being compared in Figure 4F.

In this graph, we analyze metabolite datasets from excised midbrain regions and ratio the product (Oxidized glutathione or GSSG) over the precursor (glutathione or GSH). From this analysis, we observe elevated ROS in the *Surf1* mutant mice compared to age-matched wild type litter mates.

– Label uninjured vs injured in Figure 5 E and F.

No injury was administered in Figure 5E. In this case, we are characterizing how *Surf1* mutant mice have altered metabolism prior to the injury, which our data suggests is a neuroprotective preconditioning process. Figure 5F represents all injured brains (7 days post injury). We have included additional text within the figure legend to clarify.

Writing:– In the introduction section, a paragraph summarizing your findings would be helpful in orienting the reader to the relevance of the work in relation to previous studies.

We provided the journal with a summary of our results written in lay terms. This will be submitted to *eLife* Digest.

– There are a lot of references to Parkinsons disease mediated dopaminergic neurodegeneration, but the link to the current findings is somewhat tenuous, especially as the behavioral deficits observed in the mouse injury model are only a small subset of the motor and cognitive deficits seen in Parkinsons models.

This point is well taken and we agree that additional behavioral analysis needs to be performed (preferably in the context of addiction and reward). It is noteworthy that we do recapitulate the most characteristic phenotype of the disease from a pathology standpoint (being loss of dopamine neurons). Several genetic models of Parkinson’s disease in mice do exhibit some of the behavioral phenotypes but lack the hallmark loss of dopaminergic neurons in the substantia nigra.

Reviewer #2:This study identified a neuroprotective mechanism using experimental models of blunt force trauma in *C. elegans* and concussion in mice. Interestingly, reducing Mitochondrial CIV function elevates mitochondrial-derived reactive resulted in neuroprotective effect via ROS and HIF1 mediated metabolic transition. Mechanism studies on ROS and metabolic regulation are interesting. The worm and mouse models are well established. The mouse model to reduce rather than ablate cytochrome C oxidase activity is a good choice.1. Line 124, the authors may add a few more words to describe how cox-5b is identified, as a small-scale screening has been performed.

We have included text to examine previously reported beneficial mitochondrial mutants.

2. Figure 3C-F, baseline results should be provided.

We have included baseline readings for the respective non-injured animals in the corresponding supplemental figures.

3. The authors may need to determine whether nuo-2, sdhd-1, cyc-1, atp-3 RNAi treatments reduce ROS. Not sure if ROS suppression is cox-5b RNAi specific.

Several other groups have already shown that RNAi for almost all of these ETC components increase basal ROS levels^6,7^. In the rodent, we observed a higher resting or basal ROS levels in Surf1 mutants but reduced ROS induction after trauma in the Surf1 mutant. We hypothesize that mild increases in ROS prior to injury can initiate a neuroprotective metabolic shift to glycolysis. Upon the metabolic demand of injury, cells rely less on mitochondrial respiration and do not overwhelm the ETC. Thus, we would predict that mitochondrial perturbations would increase basal levels of ROS and as long as they do not reach toxic levels, they can promote a protective metabolic switch. In this manner, it is a hermetic response. It remains to be determined whether these various ETC RNAi constructs can mitigate trauma-induced ROS fluctuations.

4. If there is a selective vulnerability of different neural subtypes, results using MEFs in figure 4 may not explain the "selective vulnerability" phenotype. Are dopaminergic neurons hypersensitive to ROS after brain injury? The mechanisms need to be better explained.

This is true regarding the MEFs, which we utilized to examine mitochondrial morphology resulting from mutating SURF1. However, we do not use the MEF cells at any point in the manuscript to assess “selective vulnerability” since no viability assays were performed on these cultured cells. The MEFs were more amenable for super-resolution analysis and provided a much clearer demonstration of how loss of SURF1 can impact mitochondrial morphology. We do include a Figure 4B showing that expression of the ROS-producing KillerRed probe exclusively in the nervous system is sufficient to kill dopaminergic neurons.

5. In figure 5, if HIF1 is the "key" regulator, does HIF1 RNAi prevent glycolytic preconditioning and neuroprotection? Similarly, does hypoxia exhibit neuroprotective effect?

We have tempered the text and replaced “key” with “important”. Due to its pleotropic nature, we were hesitant to perform the respective experiments involving the knockdown of *hif-1* expression. However, we are very interested in performing the recommended hypoxia experiments but lack the appropriate environmental chambers to perform such experiments.

Reviewer #3:Traumatic brain injury (TBI) is a significant source of global behavioral, neurological and neurodegenerative morbidity. Though risk factors for poor outcomes following TBI, such as those thought to precipitate Alzheimer's disease are intensely studies, conserved mechanisms of injury and resiliency factors with the potential to prevent poor outcomes remain largely unknown. In this study by Fonseca and colleagues, the study team leverages a combination of model systems (e.g. *C. elegans*, mice, cell systems) to identify the selective vulnerability of dopaminergic neurons following physical trauma. Through a series of well-reasoned and articulate experiments, the study team presents evidence that suppression of function at the mitochondrial electron transport chain complex IV may reduce the loss of dopaminergic neurons following TBI, in part by driving astrocyte-mediated lactate shuttling to reduce oxidative phosphorylation in vulnerable neurons. This is potentially a very significant finding with wide ranging implications not only for acute injury care but potentially chronic neurodegenerative disease as well. A clear strength of the study is its approach to identifying evolutionarily-conserved mechanisms of selective dopaminergic vulnerability to TBI, which will likely be critical to identifying actionable targets in human TBI. The study is well written with an interesting and thoughtful narrative. The weakness of the paper is primarily the under-reporting of methodological details essential to reproducing the work, inadequate controls to verify results, and a paucity of data supporting major claims and conclusions.Recommendations for the authors:Line 46: Unclear statement. Clarify.

Clarified.

Line 82: Change "degeneration" to "injury".

Text adjusted accordingly.

Line 97 to 100: TBI-induced deficits in voluntary movement are well established and may have multiple causes unrelated to SN / DA injury. Interestingly, changes in "disinhibition and risk seeking", another classical Parkinsonian trait and more cleanly associated with dopaminergic dysfunction, has been reported in TBI in human subjects (self-report) and mouse TBI models (using novel object recognition tests). Provide results of novel object recognition tests to demonstrate unencumbered behavioral / functional impairment related to dopaminergic injury / degeneration in this mouse model. Evaluate the correlation between behavioral results with the degree of dopaminergic loss across SNr, SNc, VTA, etc.

We had previously performed the respective novel object recognition studies and observed no difference 7 days after concussion between the control and injured groups. We did not report this negative data since it did not contribute to the overall story.

Line 113-115: Caution and conservativeness is warranted in this closing statement. This study has not adequately compared and contrasted the profiles of neuronal injury across the SNc/Midbrain with those of the thalamus and therefore its inappropriate to suggest that the inherent physiological properties of dopaminergic neurons sensitizes them to biomechanical insults [as compared with the thalamus]. Indeed several studies have noted substantial thalamic alterations associated with repeated TBI, whereas few have identified dopaminergic pathology.

While a majority of the TBI field has historically relied on open-head animal models, we anticipate many differences as we transition to a more physiologically relevant model of TBI, and more so concussion. To this point, allow me to highlight a large and well-controlled clinical study, which looks at over 7000 participants and accounts for over 45,000 years of clinical follow up^8^. In this study, they draw a correlation between concussion with a loss of consciousness and the progression of Parkinson’s disease, Parkinsonisms, and Lewy body dementia.

Line 139: As written, your statement makes it sound as though though evolution has selected for the protection of DA neurons against TBI by reduction of cytochrome c oxidase activity, which I do not think you intend to convey. Perhaps, "suppressing trauma-induced degeneration of dopaminergic neurons through reduction in cytochrome C oxidase activity is sufficient to conserve dopaminergic function in both *C. elegans* to mice."?

Our intension was to summarize the experimental results, which show that dopamine neurons in mice and worms are highly sensitive to trauma-induced death (when compared to other brain regions and neuronal subtypes). Thus, this process or mechanism of selective vulnerability appears to be conserved between these species.

Major concern Line 150 to 151: If I am understanding correctly… As written, worms are injured and then placed into RNAi lawns to suppress de novo translation of cox5b to the effect of mitigating ROS production. However, this does not make sense, since ROS production should be able to be generated with existing mitochondrial protein complexes and not necessarily require new protein. This would suggest the vast literature of "tired old mitochondrial proteins are the cause of ROS" is collectively wrong. Resolve this apparent conflict. Possible methods may use cycloheximide to halt general translation or use of tet on/off systems controlling target protein expression.

To resolve the confusion regarding the timeline of RNAi administration and injury, we have included additional text in the revised methods section on worm trauma. All RNAi treatments are applied to the animal throughout their life. Thus, animals have been preconditioned prior to the injury. For *cox5b* RNAi, animals have reduced translation of the COX5b protein leading up to the injury. In this case, we proposed that reduced reliance on mitochondria for energetics mitigates ROS production resulting from an overwhelmed ETC.

Line 159 to 161: Unclear / confusing. For clarity, please reword or break into separate sentences.

Adjusted text accordingly.

Line 175: Was a correlation statistically evaluated? If so, please report results. If not, change wording. Perhaps use "corresponded".

Changed text.

Line 190: "correlated". Same comment as Line 175.

Text adjusted.

Line 203: typo "Pdk2". Change to "pdk2" (standard gene symbols are all lower case by convention).

Please see our comments on standard nomenclature for each organism above.

Replace "activation" with something more precise such as "upregulation" or "increased transcription". The word activation implies signaling or physiological response.

Text was adjusted to say upregulated.

Line 204: typo "Pdp1". Change to "pdp1" (standard gene symbols are all lower case by convention).

Please see our comments on standard nomenclature for each organism above.

Line 212: Change "spatial" to "cellular".

Adjusted text.

snRNAseq provides cellular data and though cellular nuclei are physically separate from one another, snRNAseq does not preserve accepted information on spatial relationships such as that provided by microscopy.

Regarding single nuclei sequencing techniques, we do lose spatial elements upon tissue dissociation but we can pinpoint the nature of each cell during sequencing analysis by identifying select transcripts, which are specific for the respective cell type. For example, we can identify dopamine specific transcripts such as the dopamine transport, *SLC6A3* or DAT, in combination with the VMAT2 (SLC18A2) and the dopa-decarboxylase (DDC) to confidently designate these cells as dopaminergic. This new technology has become widely used and accepted within the research community. Therefore, we formed a collaboration with established experts on single cell analysis (laboratory of Dr. Konopka) to perform this respective analysis appropriately.

Line 219 to 221: "Through spatial resolution…" Identifying a pro-Warburg shift in astrocytes driven by Surf1 KO, does not demonstrate that this is a pathway for naturally occuring Warburg phenomenon. Suggest, softening by changing text to read "…may be initiated by electron…". Further, though the presented data indicates the Warburg shift is princpally manifest in astrocytes, it does not rule out the initiation of this process by extrinsic factors such as exosomes.

Unclear on this comment and the term “naturally occurring Warburg”. We observe that the transcriptional changes occur in astrocytes versus the dopamine neurons. Yet how these transcriptional changes in the astrocytes communicate to the dopamine neurons remains to be determined. Potentially, it occurs through exosomes.

Line 229: Hif1alpha… gene or protein?Line 236: Hif1alpha… gene or protein?Line 239: Hif1alpha… gene or protein?

Depending on the instance, if the word follows the gene nomenclature guidelines found below, then we intend to reference the gene. If instead, all the letters are uppercase and not italicized, then we intend to reference the protein (in accordance with the guidelines mentioned below).

- Mouse: Only the first letter uppercase, italicized (in accordance with MGI guidelines).

- Human: All letters uppercase, italicized (in accordance with HCNC guidelines).

- *C. elegans*: All letters lowercase, italicized (in accordance with WormBase guidelines).

Line 245: change "pdp-1" to "pdp1".

Worm nomenclature requires the "-".

Line 249: change to "… trauma-induced neurodegeneration of dopaminergic neurons…"Table 1.Are these parameters for mouse brain or rat brain? If they are for rat, then provide the correct parameters for mouse brain. If the title is in error, then correct the title.Provide a descriptor as to what each parameter is and what higher or lower numbers indicate for the lay reader.

In the original manuscript, parameters were defined for the rat brain. Unlike the rat brain, viscoelastic properties for some mouse brain regions (the hippocampus in particular) have not been defined or reported in the literature. To address this, we performed the respective analysis with updated mouse brain parameters with the exception of the hippocampus. In this case, we superimposed the viscoelastic parameters of the rat hippocampus into the mouse model. These new properties for the revised model have been updated in Table 1 and additional text included in the revised methods section. Regardless of the rat brain injury model or the mouse/rat hybrid model, the conclusion remains the same. TBI promotes a wave of mechanical strain, which propagates through the whole brain.

Figure 1.Panel C: Requires a reference map or annotations directly on the image to determine the precise anatomical location(s) pictured in the micrographs in order to inform the reader as to the principle cell populations involved. If this is the midbrain Substantia nigra, please label SNp, SNr, VTA, etc. If this is not precisely in that area, additional tissue staining may be required to identify cell types.

In the revised manuscript, we have included labels (1, 2 and 3) in the Figure to highlight the respective regions of the midbrain. All are included in the respective micrograph.

TH (tyrosine hydroxylase) is a marker of catecholaminergic cells and not specifically a marker of dopaminergic cells. Staining with TH will therefore include dopaminergic and noradrenergic cells which vary in their distribution throughout the midbrain to pons. If needed use dopamine β-hydroxylase (DBH) in addition to TH to differentiate dopaminergic cells from noradrenergic cells.

It is worth noting that any cell with TH will synthesize dopamine. However, noradrenergic cells convert the dopamine to norepinephrine and this occurs in the locus coeruleus, which is not in the brain section that we examined. Although noradrenergic cells invert part of the midbrain, no NA cell bodies have been reported in this brain region ^2^.

Use of cleaved caspase-3 staining is at insufficient resolution to determine whether TH cells (or an unrelated phenotype) are positive for cleaved caspase-3. Quantitation of colocalized cleaved caspase-3 across TH^+^, astrocytes, vs other cell types should be supplied.

We have provided a counter-stain in the revised manuscript with Nissl to confirm that we are assessing neurons rather than support cells.

Use of additional markers are required to verify the apoptotic cells are dopaminergic neurons and not simply other cell types such as infiltrated peripheral immune cell subsets (which are known to express tyrosine hydroxylase and apoptosize during the early phases of brain injury resolution).

We examine 28 days after injury to confirm a significant loss of TH positive cells. These numbers correspond nicely with levels of cleaved caspase observed in the same brain regions. Moreover, we do not anticipate that infiltrated peripheral immune cells would be significant contributors to overall TH-positive signal since no inflammation is observed at 7 days post injury as evidenced by RNA-seq, indirect immunofluorescence, and snRNA-seq.

Panel C: At what time point?

7 days post injury. Now included in the figure legend.

Panel D: At what time point? How many slices per animal were examined?

In the methods section of the revised manuscript, we have included additional text to provide more details regarding the confocal microscopy. Briefly, brain parenchyma images were collected as Z-stacks with 1 µm steps as follows: midbrain micrographs 20 µm/10x objective/5 tile Z-stacks, cortex micrographs 10 µm/40x objective/5 tile Z-stacks, hippocampus micrographs 20 µm/10x objective/8 tile Z-stacks, thalamus micrographs 5 µm/40x objective/2 tile Z-stacks. Number of tiles to be collected was determined by completely covering the brain structure of interest (i.e. midbrain 1x5 tiles, hippocampus 2x4 tiles). Tiled images were stitched with the Leica LAS X software. Cell culture micrographs were collected from cells grown on glass coverslips with a total of 10 fields collected with a 40x objective for image processing and analysis. Brain parenchyma images are displayed as maximal intensity projections of Z-stacks. Cell culture images are displayed as single plane micrographs.

Panel E: It is unusual for uninjured mice to fall off the rotorod so quickly, especially when they were previously trained on the task two days prior. Were the animals placed on the rod "backwards"? Provide data for all tested speeds if available.

We did not use a rotarod training paradigm. Rather we sought to perform a more naïve motor coordination analysis. In brief, mice were placed in the chamber 2 days prior to injury without turning on the machine. This time was used to familiarize the mice with their surroundings and enabled us to get more consistent readings after trauma. We have included additional text in the revised methods section to better clarify the assay and analysis.

Figure 2.Between A and B: There is an illustrated "block". I am assuming this is either an illustrated "typo" or the impactor head. Please define or remove this "block".

The block observed in Figure 2 between A and B is the depiction of the impactor seen in A, but since the frame in B is at a later timepoint, the impactor is now ascending and therefore appears further away from the brain.

C. Both Table 1 and Figure S2 indicate the finite element model includes or can include analysis of cerebellum and brainstem. Data on brainstem and cerebellum should be supplied as they are significant areas of innervation to and projections from the areas under study. Further, they are intimately involved in both righting reflex and locomotion and may lend to the interpretation of this and future studies.

In the revised manuscript, we have included analysis with the new brain regions including the brainstem and cerebellum.

Figure 3.Panel A and B: At what time point? Also, provide either quantitative real-time PCR or Western blot quanitation of relative RNAi efficiency.

All worm injury is assessed 24 hours post injury while mouse injury is assessed 7 days post-concussion unless otherwise stated. We have included the timing post-injury in the figure legend for clarity. Regarding efficiency of RNAi, we observe phenotypes upon *cox-5b* RNAi consistent with prior reports, including smaller body size and extended lifespan.

Panel C: Similar to Figure 1, provide outlines of regional anatomy and quantitation of TH vs "other cells" costaining along with "zoomed in" insets to verify colocalization of cleaved caspase-3 with dopaminergic neurons.

We have included the respective labeling system to highlight the different midbrain regions of interest. We have provided zoomed inserts of the brain micrographs in the supplemental figures (1 and 3).

Panel C: At what time point are the brain slices representing?

7 days post injury. Included in the revised figure legend.

Panel D: Are these TBI or control mice?

These are concussed mice. Very little cleaved caspase is observed in the midbrain of 12 week old mice. Please reference Figure 1 to compare control with concussion.

Panel F: Are these TBI or Control mice? If TBI, how long after TBI are the mice given this test?

Injured mice. Included in the figure legend.

Figure 4.Title: Change "Reducing cytochrome C…" to "Decreasing cytochrome C oxidase expression…". This helps avoid confusion between electrochemical "reductions" involving cytochrome C oxidase and changes in cytochrome C protein expression levels, such as the use of RNAi intends to cause.Panel A: Was there no latency between when the trauma occured and when they began peroxide measurements? If so, include that in the timescale.

Due to the nature of transferring the worms, measurements did not begin until 3 minutes post-injury. The respective time course beyond this lag is reported on the x-axis of the graph.

Panel C: Data lines are unlabeled as to the reagent used to test OCR. Graph should consist of control, actinomycin, FCCP and Oligomycin treatment conditions for each genotype. It is impossible to conclude that this even has living cells without providing more data.Recommend using a dynamic Seahorse ECAR / OCR assay. Please provide the full Seahorse Glycolytic ECAR and Mitochondrial respiratory OCR profiles.

Since we did not use a Seahorse efflux analyzer to determine OCR, no OCR profiles (which are typical outputs for the Seahorse) were obtained. Rather, we used a fluorescence commercially available kit (Cayman Chemicals) to determine OCR measurement. This and catalog numbers for the reagents are now clearly stated in the revised methods section.

Panel E: n=3. Is this technical or biological replicates? How was this measured (mention in legend)?

Biological replicates were measured with a Clariostar plate reading fluorimeter.

Figure 5.Panel B: Please run all samples on a single western blot and show results as a single boxed area. When western blots are presented such as this it is difficult to determine an effect of "injury". As it is, the reader cannot conclude that there's any difference between conditions other than a main effect of genotype.

Per the recommendation of the reviewer, we have included the whole western blot as a single boxed area in the Figure supplement.

Figure Legend Text for Panel C: remove the word "neuronal" or replace with "neural".Panel C (figure): Gene symbols are not capitalized by convention.

We have implemented the following standard gene nomenclatures:

- Mouse: Only the first letter uppercase, italicized (in accordance with MGI guidelines).

- Human: All letters uppercase, italicized (in accordance with HCNC guidelines).

- *C. elegans*: All letters lowercase, italicized (in accordance with WormBase guidelines).

Panel F: Requires single cell quantitation across DA neurons, astrocytes, others to determine treatment effects differentially affect apoptosis in a partcular target cell type.

Colocalization of cleaved caspase 3 signal corresponds nicely with TH-positive cells even at higher resolutions. GFAP positive cells were sparse within the midbrain.

Panel G and H: Change pdp-1 to pdp1.

This is not accurate nomenclature for *C. elegans.*

Figure S1.Panel C: How many slices per mouse were examined?

Included in the revised methods section.

Panel D and E: What study timepoint do these micrographs represent? How many slices per mouse were examined?

7 days post-trauma and details provided in the confocal imaging methods section of the revised manuscript.

Panel F: It is unusual to detect no change across a number of microglia enriched genes in areas that are experiencing neuronal apoptosis, especially in CD68 which would be under greater use in the elimination of cellular debris. How was this data normalized?

At 7 days post-trauma, we have not generated cranial fracture and dopaminergic neurons have not yet ruptured. Thus, it is feasible that CD68 and microglial activation has not yet occurred in this brain region. Importantly, our indirect immunofluorescence and RNA-seq data help validate our observations.

Was it library normalized per cell? If not, consider that analytical approach.

In this instance, we are performing RNA-seq on the entire brain. Normalization was based on RNA concentration.

What type of RNAseq and where was the material collected from? (I am assuming snRNAseq from the described subdissection)

In this panel we are performing RNAseq from whole brain. This information is provided in the methods section which describes the RNA isolation and library preparation as well as the model of sequencers used and the analysis methods.

Provide positive and negative controls to prove the assay worked. Ensure the microglia being reported on are from the area under study and not from a broadly mixed tissue source, which may have originally come from "uninjured" locations by sheer random chance and may not represent the SN. Microscopy or morphometric analyses may be required to ensure proximity.

Microglia were analyzed proximal to the midbrain in the revised manuscript. Based on highly sensitive RNA-seq and snRNA-seq, we have no indication that microglial activation is occurring 7 days post-concussion in our closed-head injury paradigm.

Figure S2.No comments.Figure S3.Panel A: Are these Injury or control mice?

These are non-injured mice to characterize levels of reduced cytochrome c oxidase. This is included in the revised figure legend.

Panel B: Mortality should be represented as a curve across increasing impact force. For a first demonstration of mortality curves, ideally data from at least three different cohorts, with each cohort injured separately from one another on different days, should be supplied to allow the reader to determine the degree of reproducibility for both the execution of the method and the experimental measure being made.

Good points. However, our active animal protocols for concussion limits our ability to perform the proposed experiment.

Panels D and E: Panel D shows an apparent loss of DA neurons in injured mice with a compensatory upregulation of TH by remaining neurons 28 days following injury. Panel E quantifies the DA neuron loss for both Surf wt and KO mouse strains. What is the average immunofluroescent expression of TH by remaining DA neurons after injury and does it differ by strain?

In the revised manuscript, we provide a zoomed in panel to complement the existing micrograph. However, we do not quantify TH fluorescence in individual neurons. If this did occur, we hypothesize that it may represent a compensatory mechanism to account for reduced dopamine production from this brain region.

Figure S4.No commentsFigure S5.Panel A: Are these control mice?

Yes.

Panel B to E: At what timepoint after injury does this data represent?

Metabolic cage analysis was performed for a 7-day period post-injury. Shown are averages of the respective period. This data is described in the methods section of the revised manuscript.

Panel F and G: these panels appear to be misidentified in the figure legend. Please check.

We appreciate you picking up on this. In the revised manuscript, we confirmed that figure legends match the respective figure.

Panel M: It does not seem reasonable to acheive a snRNAseq of midbrain and striatal brain tissue without obtaining microglia and endothelial cells. Please reconcile this and report these cell types.

This is a good point and exactly why we leveraged single nuclei sequencing techniques. In collaboration with established experts on single cell analysis (laboratory of Dr. Konopka), we were able to distinguish dopamine producing cells from other cell types and analyze transcriptional differences within the defining sub-groups or “clusters” via genetic profiling (Figure 5—figure supplement 1N). Since we did not observe inflammatory changes 7 days post-concussion, we excluded microglia from our analysis. Examination of brain vasculature and endothelial cells is interesting but outside the scope of the present study.

Methods:Single nuclei isolation, purification, and RNA sequencing: The reported dissection method will produce single nuclei representing subcortical / diencephalic structures, midbrain, pontine and medullary tissues. Interpretation of snRNAseq results cannot therefore be attributed to represent midbrain or striatal cell populations.

To your point, this is exactly why we used single nuclei sequencing techniques. As mentioned above, this new technique allows us to analyze transcriptional changes in select cell types from complex tissue. This is a major reason why this new technology has rapidly gained so much traction in the research community.

Closed-head traumatic brain injury: What % isoflurane, what flow rate oxygen, were the mice connected to isoflurane when impacted or not, was the scalp intact (as assumed to be but not explicitly stated)?

In the methods section of the revised manuscript, we provide all these specific details. In brief, we used 3% USP grade isoflurane and 2.5 lpm USP grade oxygen. The procedure takes under 10 seconds to set up the mice and administer the injury, so it is not necessary for the mice to remain connected to isoflurane. The scalp remained intact on these mice.

Media acidification assay: Product numbers and formulation of DMEM is critical for the interpretation of the reported method. At present the method description appears inadequate to provide the type of data desired or reported. Further, cell counts must be performed after the assay is performed since mitochondrial mutations are known to alter population growth kinetics. Therefore acidification results ought to be normalized against population numbers to control for this confound and allow an accurate interpretation of the results. Confirmation of acidification assay should also be performed using a secondary method such as pH meter.

In the revised methods section of the manuscript, we provide additional details regarding this particular assay. All data was normalized to 1x10^5^ cells for analysis. We specify the use of DMEM/F-12 (Cat. No. 11320033; Gibco), which contains phenol red with a pH indicator ranging of 6.8 to 8.2 according to the Merck Index, 13^th^ edition, 7329. Methods used in the paper were adapted from a 2018 Resource-Application note from BioTek titled, “Using phenol red to assess pH in Tissue culture media”.

Statistical Analysis: It is inadequate to state "Post-hoc analysis was determined by Prism recommendation". Please provide details on the types and specific uses of post-hoc analyses.

In the methods of the revised text, we have included more extensive details regarding our Post-hoc analysis. In brief, student’s t-test was used to compare means between two normal populations. Mann-Whitney U test was used to compare differences in the dependent variable between two groups. Post-hoc analysis performed after ANOVA included Dunnett’s multiple comparison (to compare means from several experimental groups against a control group mean) and Tukey’s multiple comparisons test (to compare all possible pairs of means). These details are provided in the revised figure legends.

RNAi administration: RNAi sequences and suppliers and concentrations must be provided.

Standardized *C. elegans* RNAi libraries (both Vidal and Ahringer) are commercially available with the respective sequence information. Please see the methods section on how RNAi cultures were prepared for experimentation.

Large-particle flow cytometry: How were data calibrated and normalized?

Prior to every run, laser power and flow rates were calibrated by use of GP control particles (COPAS Biosorter, 310-5071-001) as recommended by the company. From experiment to experiment, we maintained a consistent PTM laser power for each respective fluorophore. Moreover, we ran an empty vector control (both injured and non-injured) which we used to normalize all our experimental values.

Motor Analysis: What make, model and manufacturer is the Rotorod being reported? On the testing day, how many times were the mice tested and does the reported "latency to fall" represent the average? If it does not represent the average, what does it represent?

In the methods section of the revised manuscript, we have included the make, model, and manufacturer of our laboratory’s rotarod. Moreover, we provide a detailed description of the procedure. In brief, motor deficits in mice were measured by Rotarod (Cat. No. 76-0770; Harvard Apparatus) with a rod diameter of 3 cm and a rod height of 20 cm. Animals were not trained on a rotating rod prior to testing, but rather exposed to the Rotarod machine chamber for 2 minutes (3 times daily). For testing, animals were placed on the rod with accelerating speed (0-24 rpm in 120 s) for all experiments. Latency to fall was recorded electronically in seconds by the apparatus and values were averaged per group for reporting. The clock was stopped if an animal held to the rod on two consecutive rotations and if the animal failed to fall after 120 s. Animals were returned to their cage after each trial.

Confocal imaging: Was a single image plane acquired or were they z-stack images? How were the images controled between subjects / slices? If they were z-stacks, what was the step size? How were the images displayed in the manuscript- were they single planes or z-stacks? Were all the image volumes identical?

In the methods section of the revised manuscript, we have included additional text to provide more details regarding the confocal microscopy. Briefly, brain parenchyma images were collected as Z-stacks with 1 µm steps as follows: midbrain micrographs 20 µm/10x objective/5 tile Z-stacks, cortex micrographs 10 µm/40x objective/5 tile Z-stacks, hippocampus micrographs 20 µm/10x objective/8 tile Z-stacks, thalamus micrographs 5 µm/40x objective/2 tile Z-stacks. Number of tiles to be collected was determined by completely covering the brain structure of interest (i.e. midbrain 1x5 tiles, hippocampus 2x4 tiles). Tiled images were stitched with the Leica LAS X software. Cell culture micrographs were collected from cells grown on glass coverslips with a total of 10 fields collected with a 40x objective for image processing and analysis. Brain parenchyma images are displayed as maximal intensity projections of Z-stacks. Cell culture images are displayed as single plane micrographs.

Protein extraction and western blotting: Reported methods do not appear to include the use of a phosphatase inhibitor. Therefore western blots showing any changes in phosphorylation may be related more to target protein abundance and not differential phosphorylation. Westerns blots therefore ought to include probes against the non-phosphorylated "total" target protein then in order to help interpret apparent expression changes.

All cell extracts were created in the presence of the SDS detergent, which will linearize folded proteins and inactivate enzymes including phosphatases and kinases. Even alkaline phosphatases, which are known to be resistant to lower concentrations of SDS treatments, can be inactivated with 1% SDS^5^, which is the concentration used in our lysis buffer. This point has been added to the revised methods section. Regardless, we have now included additional western blots of total PDHE1 and confirm that its steady-state levels are not significantly altered in the different conditions.